# Hydrodynamic simulation of the effects of stable in-channel large wood on the flood hydrographs of a low mountain range creek, Ore Mountains, Germany

Daniel Rasche[1,2], Christian Reinhardt-Imjela[1], Achim Schulte[1], Robert Wenzel[1,3]

[1]Freie Universität Berlin, Department of Earth Sciences, Institute of Geographical Sciences, Berlin, 12249, Germany
[2]GFZ German Research Centre for Geosciences, Section Hydrology, Potsdam, 14473, Germany
[3]LfU State Office of the Environment Brandenburg, Potsdam, 14476, Germany

*Correspondence to*: Christian Reinhardt-Imjela (christian.reinhardt-imjela@fu-berlin.de)

**Abstract.** Large wood (LW) can alter the hydromorphological and hydraulic characteristics of rivers and streams and may act positively on a river's ecology by i.e. leading to an increased habitat availability. On the contrary, floating as well as stable LW is a potential threat for anthropogenic goods and infrastructure during flood events. Concerning the contradiction of potential risks as well as positive ecological impacts, addressing the physical effects of stable large wood is highly important. Hydrodynamic models offer the possibility of investigating the hydraulic effects of anchored large wood. However, the work- and time-consumption varies between approaches of incorporating large wood in hydrodynamic models. In this study, a two-dimensional hydraulic model is set up for a mountain creek to simulate the hydraulic effects of stable LW and to compare multiple methods to account for large wood induced roughness. LW is implemented by changing in-channel roughness coefficients and by adding topographic elements to the model in order to determine which method most accurately simulates observed hydrographs and to provide guidance for future hydrodynamic modelling of stable large wood with two-dimensional models.

The study area comprises a 282 m long reach of the Ullersdorfer Teichbächel, a creek in the Ore Mountains (south-eastern Germany). Discharge time series from field experiments allow a validation of the model outputs with field observations with and without stable LW. We iterate in-channel roughness coefficients to best fit the mean simulated and observed flood hydrographs with and without LW at the downstream reach outlet. As an alternative approach of modelling LW induced effects, we use simplified discrete topographic elements representing individual LW elements in the channel.

In general, the simulations reveal a high goodness-of-fit of between the observed flood hydrographs and the model results without and with stable in-channel large wood. The best fit of simulation and mean observed hydrograph with in-channel LW can be obtained when increasing in-channel roughness coefficients in the entire reach instead of an increase at LW positions only. The best fit in terms of the hydrograph's general shape can be achieved by integrating discrete elements into the calculation mesh. The results illustrate that the mean observed hydrograph can be satisfactorily modelled using an adjustment of roughness coefficients.

In conclusion, a time-consuming and work-intensive mesh manipulation is suitable for analysing more detailed effects of stable LW on small spatio-temporal scale where high precision is required. In contrast, the reach-wise adjustment of in-channel roughness coefficients suggests to provide similarly accurate results on the reach-scale and thus, can be helpful for practical applications of model-based impact assessments of stable large wood on flood hydrographs of small streams and rivers.

**1 Introduction**

Large wood (LW) is a natural structural element of rivers and streams with forested catchments (Gurnell et al., 2002; Roni et al., 2015). It is part of the permanently produced amount of plantal detritus in terrestrial ecosystems before it enters rivers and surrounding riparian areas (Wohl, 2015). In fluvial systems, large wood can be defined as dead organic matter with woody texture, having diameters of at least 0.1 m (Kail and Gerhard, 2003). Unlike in the definition of Kail and Gerhard (2003),

several studies include the length of large wood of at least 1 m for distinction (i.e. Gurnell et al., 2002; Andreoli et al., 2007; Comiti et al., 2008; Bocchiola, 2011; Kramer and Wohl, 2017; Wohl, 2017). The latter definition is adapted in the present study.

Large wood influences the physical structure of watercourses as it increases streambed heterogeneity by forming scour pools (Abbe and Montogomery, 1996), causing sediment sorting and altering water depth as well as flow velocity (Pilotto et al.,

2014). Hence, the presence of large wood can lead to increased habitat availability in rivers and streams (Wohl, 2017). Positive ecological impacts of LW on fish species (i.e. Kail et al., 2007; Roni et al., 2015) and the macro-invertebrate fauna (i.e. Seidel and Mutz, 2012; Pilotto et al., 2014; Roni et al., 2015) are documented. A recent review of the hydromorphological and ecological effects of LW with focus on river restoration can be found in Grabowski et al. (2019).

In stream restoration projects, the presence of large wood can result in rapid hydromorphological improvements (Kail et al.,

2007). Consequently, wood placements have a high potential for stream restoration measures (Kail and Hering, 2005); for instance, in Germany, where many watercourses lack of a high hydromorphological diversity (BMUB/UBA, 2016).

Large wood assemblages and elements are more likely to be stable when their length exceeds channel width (i.e. Gurnell et al., 2002), most likely to occur in small first order streams and rivers, which in turn are the most abundant order of water courses on the planet (Downing et al., 2012). However, even in small but steep headwater streams, large wood may be

transported during hydrogeomorphic events of high magnitude such as debris flows (Galia et al., 2018) or extreme floods. A conceptual model for a first estimate of large wood transport in water courses is given in Kramer and Wohl (2017) including hydrological as well as morphological variables. Further detailed information about large wood dynamics in river networks can be found in recent reviews of Ruiz-Villanueva et al. (2016a) and Wohl (2017). Large wood may drift during floods, elements jam at bridges or other infrastructure and cause increased water levels, damage or completely destroy anthropogenic

goods and structures (Schmocker and Hager, 2011). On the contrary, stable large wood reduces water conveyance (Wenzel et al., 2014) and leads to increased water levels upstream and in turn, increased risk of flooding and water logging in surrounding areas. For these reasons, LW is removed from European rivers and streams for more than a century (Wohl, 2015) also to ensure

navigability in larger rivers (Young, 1991). As a result, the usage of LW in river restoration in the form of leaving naturally transported wood in-stream or artificial stable wood placements is discussed controversially (Roni et al., 2015; Wohl, 2017). With respect to the potential risks of large wood for anthropogenic goods on the one hand and high ecological benefits on the other, it may be necessary to distinguish river sections in which large wood can remain or be introduced from those where it needs to be removed (Wohl, 2017). Large wood related segmentation of rivers and streams requires knowledge of the physical effects caused by mobile and stable in-channel large wood. Although several studies address the general hydraulic impact of LW in field studies (i.e. Daniels and Rhoads, 2004; Daniels and Rhoads, 2007; Wenzel et al., 2014), laboratory experiments (i.e. Young, 1991; Davidson and Eaton, 2013; Bennett et al., 2015) and reviews (i.e., Gippel, 1995; Montgomery et al., 2003) regarding the alteration of water level, flow pattern, flow velocity and discharge, a project and site specific examination is necessary to evaluate local consequences of intended stream restoration measures.

The resulting physical effects of stable in-channel LW (Smith et al. 2011) as well as the mobility, transport and deposition of large wood (i.e. Ruiz-Villanueva et al., 2014; Ruiz-Villanueva et al., 2016b) can be addressed using numerical hydrodynamic models. Numerical hydrodynamic models for the simulation of open-channel hydraulics can be classified by their dimension and solve the shallow water equations (SWE) in their one-, two- or three-dimensional form for simulating channel flow in just one (x-)direction (1D), horizontally resolved (x- and y-direction) but depth-averaged (2D) or fully resolved in x-, y- and, z-direction (Liu, 2014). Due to i.e. the increasing effort of work and computational time with increasing dimension, the applicability of 1D, 2D or 3D models depends on the scale and phenomena of interest (Liu, 2014). For simulating the general hydraulic behaviour on reach-scale, 2D models are useful tools (Liu, 2014). A detailed description of the different model types and examples of application can be found in Liu (2014) or Tonina and Jorde (2013) with focus on ecohydraulics. Several studies consider stable large wood in the scope of one- and two-dimensional hydrodynamic simulations for example for investigating its influence on flood hydrographs (Thomas and Nisbet, 2012), on floodplain connectivity (Keys et al., 2018) or are considered in research applications with an ecological focus by investigating the effect of stable LW on habitat availability or suitability (i.e. He et al., 2009; Hafs et al., 2014). In addition, Lange et al. (2015) simulate the effect of roughness elements including stable LW in the scope of stream restoration analyses. Regarding the hydraulic impact of stable large wood on flood hydrographs, Thomas and Nisbet (2012) simulate large wood to delay flood passage but no attenuation of peak discharge is modelled. Similar effects of stable LW on flood hydrographs were investigated by Wenzel et al. (2014) in field experiments, where a delay and a narrower shape through a transformation from higher to lower discharges, but only a minor attenuation of the average flood hydrograph was observed. Furthermore, representing and integrating large wood elements in hydrodynamic models is addressed in different studies using three-dimensional hydrodynamic models (i.e. Smith et al., 2011; Allen and Smith, 2012; Lai and Bandrowski, 2014; Xu and Liu, 2017). However, the modelling approach applied varies with studies. As an extensive review of applicable numerical hydrodynamic modelling systems and approaches for simulating large wood is beyond the scope of the present study, a recent overview with focus on LW dynamics as well as the representation of large wood and vegetation in simulations can be found in Bertoldi and Ruiz-Villanueva (2017).

Despite the necessity of a discrete representation of stable large wood elements in the calculation mesh of hydrodynamic models for obtaining accurate results (Smith et al., 2011) and as conducted in different studies (i.e. Hafs et al., 2014; Lange et al., 2015; Keys et al., 2018), LW elements are often accounted for using roughness coefficients in hydrodynamic model applications (Smith et al., 2011). The impact of large wood on in-channel roughness is investigated by Gregory et al. (1985),

Shields and Smith (1992), Shields and Gippel (1995), Dudley et al. (1998), MacFarlane and Wohl, (2003) and Wilcox and Wohl (2006). In addition, Curran and Wohl (2003) and Wilcox et al. (2006) have studied its partial contribution to channel roughness coefficients. However, a methodological lack remains in quantitatively estimating LW related changes of in-channel roughness coefficients (Wohl, 2017), especially under field conditions (Wilcox et al., 2006). The large wood induced alteration of channel roughness coefficients and overall hydraulic impacts such as backwater effects, for instance investigated by Schalko

et al. (2019), are crucial for the identification of local risks. Therefore, remaining knowledge gaps in these fields lead to uncertainties regarding the use large wood in river restoration and natural flood risk management in practice (Grabowski et al., 2019) and may hamper its application. Against this background, the aim of the present study is to simulate the physical effects of stable in-channel LW elements on flood hydrographs in a creek reach in low mountain ranges using a two-dimensional hydrodynamic model and previously conducted field experiments, explicitly described in Wenzel et al. (2014).

The field data offer the rare opportunity to validate simulated large wood related hydraulic effects on hydrographs of small flood events. By conducting different hydrodynamic simulations, we aim (1) for the quantification of the change of channel roughness coefficients in the entire channel or at LW positions, necessary to obtain most accurate model results of flood hydrographs with stable large wood elements in the channel. As discrete LW elements are required for most accurate model results (Smith et al. 2011), we aim (2) for comparing previous model results with simulations with discrete large wood elements

created through manipulating the calculation mesh. However, the integration of discrete elements into the calculation mesh can be highly time- and work-intensive (Lai and Bandrowski, 2014), which becomes especially true for larger scale applications. Hence, a comparison of the simulation accuracy between incorporating large wood through a rather quick change of channel roughness coefficients and as time-demanding simplified mesh elements can be provide beneficial information for future studies simulating stable large wood related effects on stream hydraulics and ecology.

Although limited to smaller streams and rivers where large wood jams and elements can be assumed as stable or situations in which large wood elements are anchored, the present study can contribute to the ability of predicting hydraulic impacts of stable in-channel large wood within hydrodynamic simulations and can also provide beneficial practical information for conducting simulation-based impact assessments of stream restoration projects considering stable large wood by comparing different methods of large wood roughness modelling.

**2 Study reach**

The study reach comprises a 282 m long section of the Ullersdorfer Teichbächel, a small first order headwater creek located in the Ore Mountains, south-eastern Germany. The catchment of the Ullersdorfer Teichbächel (50°36'48.52" N, 13°15'51.24"

E, WGS84) covers an area of 1.8 km² and drains into the river Elbe via several higher order tributaries including Schwarze Pockau, Flöha, Zschopau and Mulde.

The study reach is located in the catchment's centre and approximately 50 m downstream of an artificial rafting pond built in the 16[th] century. Two Thomson-weirs mark the study reach's upper and lower limits at elevations of 754.1 and 744.5 a.s.l. (Fig. 1) resulting in a difference in elevation of 10.4 m and an average channel gradient of 3.7 %. Channel dimensions vary strongly along the study reach, i.e. channel width ranges from < 0.8 to 2 m. Similarly, a high variability of stream bed grain sizes can be detected (Fig. 2). Moderately steep sections with a sand and fine gravel dominated bed structure alternate with reach sections of higher gradients dominated by coarse gravel, cobbles and small boulders with sizes of up to 0.3 m in diameter. The boulders consist of gneiss varieties representing the dominating bed rock formations in the catchment. Beside a highly variable stream width, alternating slope gradients and grain sizes lead to an alternation of stream depth along the study reach and hence, a generally complex channel structure.

The overall morphological character along the 282 m study reach consists of riffle-pool sequences in moderately steep sections as well as step-pool morphologies along sections having smaller channel widths and larger in-channel boulders (Fig. 2). In the latter, channel-spanning steps with corresponding hydraulic jumps and eroded pools have been observed in May 2017.

The majority of the catchment of the Ullersdorfer Teichbächel is covered with coniferous forest on prevailing cambisols and podzols including scattered deciduous trees sprinkled in. The dominating species is spruce (*Picea abies*) with occasional occurrence of mountain pines (*Pinus mugo*) and beech trees (*Fagus sylvatica*) (Wenzel et al., 2014). Trees occur only scatteredly in the narrow floodplain along the channel of the study reach with grassy vegetation on fluvic gleysols covering most parts. However, smaller floodplain sections are covered with bare soil or leaf litter. Perpendicular to the direction of flow, the maximum width of the floodplain measured from channel banks varies between 7 and 0 m, when channel banks immediately change into the embankments confining the study reach.

At the nearest gauging station Zöblitz, which is located approximately 13 km downstream the catchment's outlet at the river Schwarze Pockau and drains an area of 125 km², the mean annual discharge is 2.29 m³ s$^{-1}$. If the value is extrapolated using a regional analysis based on drainage areas, the mean discharge at the outlet of the study reach is 16 l s$^{-1}$. The flow regime of the study area is dominated by snow melt generating high flows in March and April (gauge Zöblitz, period 1937 to 2015; LfULG, 2017a). Floods of low to medium magnitudes are generated by intense snowmelt and rainfall on snow in spring or by storm events in summer. Larger flood events are caused by summer storms only (Petrow et al., 2007) but the flood magnitudes are strongly influenced by land use and are greatly affected by past forest changes (Reinhardt-Imjela et al., 2018).

## 3 Material and methods

### 3.1 The hydrodynamic model HYDRO_AS-2D

In this study, the two-dimensional hydrodynamic model HYDRO_AS-2D (version 2.2) is used to simulate the flow in the study reach with and without LW. HYDRO_AS-2D was developed for practical applications in water management (Nujić,

2006) and is used in several studies simulating flow conditions in river sections for flood risk management (i.e. Rieger and Disse, 2013) or with an ecological focus (i.e. Lange et al., 2015) and can produce a higher goodness-of-fit compared to other two-dimensional models as exemplarily shown in Lavoie and Mahdi (2017). Especially in southern Germany and Austria, HYDRO_AS-2D became a standard 2D modelling system for hydrodynamic model applications (Faber et al., 2012). Due to the numerical approaches used in the modelling system, HYDRO_AS-2D is capable of simulating mass exchange between channel and forelands, streams comprising hydraulic jumps, steep channel sections and a high variability of channel width as well as dike breaches (Nujić, 2006). The latter is to some extent comparable with the rapid release of water initiated by opening the flap gate weir used in the field experiments (see chapter 3.2). For the above-named reasons, HYDRO_AS-2D was chosen for the present study.

HYDRO_AS-2D solves the two-dimensional shallow water equations (SWE) at each node of a linear calculation mesh composed of quadrilateral and triangular elements of different sizes, representing a digital terrain model of the channel and the forelands. Shallow water equations are solved using finite volume approximations for spatial discretion, while time is discretized using second order Runge-Kutta methods (Nujić, 2006). Water flow is computed through all sides of the control volume around each node using different order polynomials and upwind schemes (Nujić, 2006). Surface roughness is represented by Strickler coefficients defined for each element of the calculation mesh. Similarly, local viscosity can be defined for each mesh element. Mesh generation, pre-processing, the setting of model boundary conditions as well as simulation result visualisation of HYDRO_AS-2D v2.2 is conducted using the software Surface Water Modelling System (SMS) v10.1 (Aquaveo Inc., USA). An overview of the methodological procedure described in the following sections can be found in Ffigure 3.

## 3.2 Datasets and mesh generation

The presented study is based on data previously collected during field experiments in March 2008 (Wenzel et al., 2014) in the river section under investigation. In this earlier study, the pond upstream the experimental reach was dammed using a flap gate weir and multiple flood waves of equal magnitude (return period of 3.5 years) were generated. The first 8 experimental runs were conducted with 9 large wood elements (spruce tree tops with a length ranging from 3 to 11.5 m, mean length 8.5 m), which were placed and fastened in the channel lengthwise 9 months earlier. After the experimental runs with LW, all LW elements were removed and 12 additional flood waves were generated without the trees. During all experimental runs, water levels were continuously recorded with a temporal resolution of 1 s at the beginning and end of the river section using Thomson-weirs equipped with pressure gauges. For each Thomson-weir, the averaged (mean) hydrograph of experimental runs with and without LW is calculated and used as the upper model boundary condition (Thomson-weir 1) and for the validation of model outputs (lower boundary condition, Thomson-weir 2), respectively.

During the development of the hydraulic model, a measurement error was detected in the water level measurement at Thomson-weir 1 (input weir), which results in a significantly lower discharge volume at Thomson-weir 2, although larger water inflows between both weirs were not observed in the field. The measurement error of the input weir was corrected by increasing water

levels in the original water level time series of the pressure gauge and recalculating discharge. The measured water levels at the first weir had to be increased by a maximum of 0.024 m until the total flood volume at both weirs was nearly equal (Tab. 1).

To generate a digital terrain model (DTM) for the studied river section, data from a cross-sectional geodetic survey conducted
with a Spectra Precision AB Geodimeter 400 in 2008 were available. To improve the implementation of the channel in the hydrodynamic model the channel width was surveyed again in intervals of 5 m using a measuring stick in May 2017. Furthermore, a digital elevation model with a spatial resolution of 2 x 2 m (Saxon State Office of Geoinformation and Surveying, 2008) is used for better reproduction of the floodplain morphology. The final DTM for the model is generated from processing and combining all topographic datasets in the software environment ArcGIS v10.5 (ESRI Inc., USA) creating a
triangular irregular network (TIN) before transforming it into a raster dataset. The resulting DTM is exported as equally spaced elevation points with a spatial resolution of approximately 0.5 x 0.5 m for the entire study reach including riparian areas and embankments. From the point grid the calculation mesh required for simulations with HYDRO_AS-2D is created. Mesh generation is done in the software environment SMS v10.1 and according to mesh quality requirements of HYDRO_AS-2D, such as minimum and maximum angle of mesh elements or maximum number of element connections per node (Nujić, 2006).
The calculation mesh is composed of quadrilateral and triangular elements. In the channel of the study reach, quadrilateral elements are created by stepwise mesh generation between cross-sectional point elevation profiles through linear interpolation of elevation between profiles. A triangular mesh is generated in the riparian areas and along embankments by using equally spaced elevation points. After merging quadrilateral channel elements and triangular foreland elements as well as including additional topographic features to the calculation mesh (Fig. 4) to match field observations, roughness coefficients are assigned
to each mesh element. The Strickler coefficients $k_{st}$ were estimated for channel sections with similar bed material and the floodplain during field surveys in May 2017 with reference to established roughness coefficient classifications for different land cover and surface material types (i.e. Chow, 1959) as well as in accordance with observed ground cover during field experiments in 2008.

### 3.3 Hydrodynamic modelling

Boundary conditions for the unsteady hydrodynamic simulations are defined in SMS v10.1. For flow simulations of the experimental reach without LW, the averaged discharge time series without LW at Thomson-weir 1 (Fig. 5) is defined as the water inflow into the study reach. Water influx is defined at the location of Thomson-weir 1 in the calculation mesh, represented by the uppermost cross-sectional nodestring in the channel. For the simulations with in-channel LW, the averaged time series with LW at Thomson-weir 1 (Fig. 5) is used as the system input.

For the simulations without and with LW, the inflow hydrographs at Thomson-weir 1 are extended forwardly by 5400 seconds using the first discharge value of the corrected mean experimental hydrograph without and with LW. This is done to achieve field conditions of minor flow through the channel in the study reach before the experimental flood waves enter the channel.

This results in a total simulation time of 9000 seconds for each simulation with and without LW with a temporal resolution of 1 second.

Simulation results are obtained at the location of Thomson-weir 2 in the calculation mesh represented by the lowermost cross-sectional nodestring in the channel of the study reach. Model performance is assessed by visual comparison of mean observed and simulated flood hydrographs without and with LW at Thomson-weir 2 as well as by calculating the statistical goodness-of-fit parameters Nash-Sutcliffe-Efficiency (NSE), percent bias (PBIAS) and RSR (ratio of the root mean square error to the standard deviation of observed values) using the hydroGOF package by Zambrano-Bigiarini (2017) in R (R Core Team, 2017). For NSE a value of 1 indicates the highest model accuracy while the optimum value for RSR and PBIAS is 0 (Moriasi et al., 2007).

## 3.4 Hydrodynamic simulation variants

In the scope of this study, four different simulation variants are applied to investigate effects of in-channel large wood on flood hydrographs in a small low mountain stream: (1) the reference variant RV representing the simulation of field experiments without in-channel LW and (2-4) variants V1 to V3 for simulating field experiments with LW.

**Variant RV** is used to obtain the best fit of the mean observed and simulated hydrograph without LW at Thomson-weir 2 through iteratively adjusting Strickler roughness coefficients in the channel and in riparian areas. In the reference variant and all other simulation variants calibration is performed to achieve the best possible simulation of the moment of rise, the rising limb and peak discharge of the mean observed hydrograph at Thomson-weir 2. Calibrated roughness coefficients leading to the best fit in variant RV will be used as initial roughness coefficients in the calculation mesh of variants V1, V2 and V3.

**Variant V1** represents the first simulation with LW. Calibrated Strickler coefficients from variant RV are iteratively adjusted for the entire channel (integrated roughness). Adjustments are made percent-wise and with equal magnitude to enable equal scaling of spatially varying roughness coefficients of mesh elements in the channel. This approach was included because the integrated channel roughness of a river section is an important input parameter for rainfall-runoff models at mesoscale or of larger watersheds, which often use only one Strickler (or Manning) coefficient per section.

Similarly, roughness is scaled in **variant V2**, in which Strickler coefficients from variant RV are adjusted at the positions of all LW elements only. LW element locations and corresponding LW influenced channel sections (length of each LW element) are derived from Wenzel et al. (2014). For each channel section roughness coefficients are adjusted percent-wise and with equal magnitude.

In contrast to variants V1 and V2, where LW is represented by reach-wise and section-wise adjustment of Strickler coefficients of quadrilateral in-channel calculation mesh elements, **variant V3** includes the integration of simplified discrete roughness elements by manipulating the existing calculation mesh used in variant RV. Therefore, discrete elements with the maximum stem length and width (without branches) of each individual LW are incorporated into the calculation mesh by creating corresponding rectangular polygons overlying the mesh. Polygons are positioned in order to have the largest possible part located in the channel of the study reach. Based on the existing calculation mesh, new mesh nodes are positioned in 0.2 m

intervals along polygon boundaries and within a 0.1 m distance outside polygons. Nodes along polygon boundaries receive the elevation of the closest upstream node increased by 1.5 m. The elevation of nodes within 0.1 m distance is interpolated from the existing calculation mesh. As mesh quality requirements (see chapter 3.2) need to be maintained, positions of some added nodes are slightly shifted. Additional quadrilateral and triangular mesh elements are created between nodes added to the mesh.

All newly created mesh elements representing discrete LW elements (Fig. 4) are parameterized with the same Strickler coefficient in order to retrieve the best fit between simulated and mean observed hydrograph with LW at Thomson-weir 2. Strickler coefficients of mesh elements representing discrete large wood elements are used to account for i.e. branches of real spruce tree tops implemented into the channel during the field experiments. Coefficients are determined iteratively during calibration of simulation variant V3.

## 4 Results

### 4.1 Simulation variant RV

In the reference variant, the best fit in the unsteady hydrodynamic simulation without LW was achieved with in-channel Strickler coefficients ranging from 6 $m^{1/3}$ $s^{-1}$ for channel sections with larger boulders to 12 $m^{1/3}$ $s^{-1}$ in channel sections where fine gravel forms the stream bed. A Strickler coefficient of 3.5 $m^{1/3}$ $s^{-1}$ was defined for riparian areas during calibration. The

distribution of calibrated Strickler coefficients in the study reach of all simulation variants can be found in ~~f~~Fig.~~ure~~ 6. Observed and simulated hydrographs of the simulation are shown in ~~f~~Fig.~~ure~~ 7. In general, the model closely simulates the characteristics of the observed hydrograph. Only the crest is slightly wider in the model and a slight model underestimation can be observed at the beginning and in the second half of the simulation time. The good model performance is reflected by a high NSE of 0.99 as well as a low RSR (0.11) and PBIAS (-3.5 %). The statistical goodness-of-fit parameters of all simulation

variants are summarized in ~~T~~table 2. The cumulative maximum inundated area comprises 739 m², defined as the total area of mesh elements inundated during simulation.

### 4.2 Variant V1 - Integrated increase of roughness in the channel

In the first simulation variant V1 of field experiments with in-channel large wood, Strickler coefficients were decreased in the entire channel based on the coefficients of the simulation without large wood (variant RV). A decrease of Strickler values and hence,

an increase of roughness of 30 % in the entire channel resulted in the best fit between the mean observed and simulated hydrograph. Consequently, in-channel Strickler coefficients range from 4.2 to 8.4 $m^{1/3}$ $s^{-1}$ in variant V1 (~~F~~fig. 6). The 9 LW elements in the field investigations cover 75.1 m of the 282 m long channel reach, i.e. the simulated 30 % increase of the integrated channel roughness refers to a LW percentage of 27 % of the channel length.

The resulting simulated hydrograph of variant V1 shows a good representation of the time of rise as well as the rising limb of

the observed hydrograph (Fig. 7). However, in the peak discharge phase the simulated hydrograph does not rise continuously until peak values are reached. If the Strickler coefficients in the channel foreland (riparian area) were decreased from 3.5 to

2.4 $m^{1/3}$ $s^{-1}$ in addition to the channel roughness, the break in the crest of the hydrograph disappears (see chapter 5.2). After adjusting roughness coefficients in riparian areas, rising limb and peak phase of the observed hydrograph are represented slightly better. Nevertheless, discharge values during peak phase show a distinct underestimation of observed values. Similarly, differences can be found along the falling limb between observation and simulation. The maximum inundated area comprises 861 m² before and 880 m² after riparian roughness adjustment. Nash-Sutcliffe-Efficiency values of 0.97 before and 0.98 after adjustment of roughness coefficients in riparian areas were achieved. The RSR shows values of 0.18 and 0.14 before and after adjustment, while PBIAS slightly increases after adjustment from -3.6 to -3.7 %. (Table 2).

### 4.3 Variant V2 - Increase of roughness in LW sections

In simulation variant V2, in-channel roughness coefficient derived from variant RV were altered in large wood affected channel sections only. Here, a reduction of Strickler coefficients of 55 % resulted in the best fit of observed and simulated hydrographs. Depending on the LW affected channel section, Strickler coefficients between 3.6 and 5.4 $m^{1/3}$ $s^{-1}$ were derived (Ffig. 6). The resulting simulated hydrograph properly represents the time of rise. Compared to variant V1, the rising limb is less accurately modelled. Similarly, to variant V1, a discontinuous peak phase is generated in the simulations. Again, an increase of the roughness in riparian areas is necessary to simulate a hydrograph with a more realistic, continuous rise of discharge up to the crest of the hydrograph. Strickler coefficients in riparian forelands were reduced from 3.5 to 1.9 $m^{1/3}$ $s^{-1}$. In addition, both simulated hydrographs (with and without subsequent adjustment of riparian roughness coefficients) show an overestimation of the observed discharge along the falling limb of the flood wave, while a distinct underestimation can be observed during the peak phase as well as in the beginning and the end of the experiments (Fig. 7). Before adjusting riparian surface roughness, the maximum cumulative inundation area is 859 m². After subsequent adjustment inundated area rises to 892 m². NSE values range from 0.94 before to 0.96 after adjusting riparian Strickler coefficients, while RSR decreased from 0.24 to 0.19 and PBIAS from -4.2 to -4.0 (Table 2). With regard to the general shape of simulated hydrographs as well as the statistical model performance assessment, variant V1 reveals a better representation of the observed hydrograph of the field experiments with in-channel large wood.

### 4.4 Variant V3 - Implementation of LW as discrete elements

In the last simulation variant (V3), large wood is integrated into the model as simplified discrete elements by manipulating the calculation mesh. The created mesh elements representing discrete LW elements received a Strickler coefficient of 8.5 $m^{1/3}$ $s^{-1}$ to account for branches and in order to obtain the best fit between mean observed and simulated hydrograph (fFig. 6). As shown in fig Figure 7, the simulated hydrograph rises slightly later than the mean observed hydrograph, which results in differences between simulation and observation along the falling limb. Additionally, a slight overestimation of peak discharges can be observed as well as the underestimation of discharges in the beginning and end of the simulation. The maximum water covered area comprises 927 m² and is much larger than in previous simulation variants. Statistical goodness-of-fit parameters show an NSE value of 0.90, a RSR value of 0.32 and PBIAS of -7.7 %. Especially the PBIAS of variant V3 is much higher

than in all other simulation variants (Table 2). According to the classification of Moriasi et al. (2007), goodness-of-fit parameter values calculated for variant V3 as well as for all other simulation variants in this study indicate simulation results of high accuracy. Despite the temporal shift between the average simulated and observed flood hydrograph as well as the lower goodness-of-fit according to the classification of Moriasi et al. (2007), the general narrow shape of the flood hydrograph of

the field experiments with in-channel LW is most accurately modelled in variant V3.

## 5 Discussion

### 5.1 Simulations of flood hydrographs in the investigated creek section

In general, the 2D hydrodynamic model closely mimics the flow conditions of the field experiments without LW (variant RV). Especially the time of rise, the rising limb and the flood peak are accurately represented, minor deviations can be observed

along the hydrograph's falling limb only due to the broader shape of the simulated hydrograph. However, it has to be noted that measurement errors may also occur in the field data demonstrated by the fact that the input time series measured at Thomson weir-1 had to be corrected to reduce the volume error between both weirs. After the correction, the cumulated volume error between both weirs was reduced to 4 m³ h⁻¹ without LW and 5 m³ h⁻¹ for the field experiments with LW (1 l s⁻¹) (Table 1). The remaining difference between both weirs lies in the range of what can be estimated as natural water influx between

both weirs based on runoff per km² estimations from regional analyses of the nearest gauging station for the days of the field experiments (LfULG, 2017b). Depending on the spatial resolution of the DTM used for calculation (2 and 5 m), the average water influx ranges from 3 to 6 m³ h⁻¹. Hence, the remaining volumetric difference can be attributed to diffuse lateral water influx during the run time of each experiment and are likely to be responsible for the modelled (Fig. 7) and observed (Fig. 5) lower discharges before and after flood passage at Thomson-weir 2. However, after correction it can be assumed that the

measured data are a reliable reference for the hydrodynamic simulation.

The broader shape of the simulated hydrograph is likely to be caused by the calculation mesh used, representing the terrain surface. The calculation mesh is based on topographic field data gathered in the scope of the field experiments in 2008 to find most suitable locations to position large wood elements (Wenzel et al., 2014). Therefore, small topographic features in the channel and adjacent riparian areas are not included in the elevation data set and hence, in the calculation mesh. This especially

applies to step-pool sequences in the study reach. Steps and pools produce rapid flow energy losses caused by corresponding hydraulic jumps and result in a deceleration of flow (Wilcox et al., 2011), where the amount of energy loss dynamically depends on water level (Comiti et al., 2009). Furthermore, erosion and transport of bed material leads to flow energy losses (Yen, 2002). As such features are missing in the calculation mesh, roughness coefficients are used to account for their impact on water flow. However, calibrating in-channel roughness coefficients may lead to a much more continuous decrease of flow

velocities instead of intense, punctual flow decelerations with implications for downstream flow conditions, in turn resulting in a broader peak of the simulated flood hydrograph. This illustrates the necessity of a high-resolution calculation mesh

including small scale topographic features in the channel and microtopography in riparian areas to obtain accurate model results.

Despite the discrepancies described above, the simulation of variant RV shows a very precise simulation of the observed hydrograph of the field experiments without large wood, which is also indicated by the statistical goodness-of-fit parameters revealing a very high model accuracy according to the classification of Moriasi et al. (2007). Hence, averaged flood hydrographs of the field experiments without large wood can be accurately simulated using the set-up model, illustrating its applicability for simulating the flow conditions in the study reach.

## 5.2 Simulating the hydraulic impact of stable in-channel LW using roughness coefficients

In simulation variants V1 and V2, roughness coefficients are used to represent large wood in the study reach. Both variants show a correct simulation of the time of rise of the flood hydrograph. Differences occur along the rising limb as well as the hydrograph's peak. Here, variant V1 produces a better fitting hydrograph. Compared to the simulation result of the mean observed hydrograph of the field experiments without in-channel LW, variants V1 and V2 produce less closely fitting simulated hydrographs, which is also indicated by the slightly lower values of statistical goodness-of-fit parameters. Nevertheless, these values still indicate a very high model accuracy, suggesting that a less time-consuming adjustment of roughness coefficients allows an accurate simulation of stable large wood induced hydraulic effects.

In-channel LW elements decelerate flow beyond their own dimensions by generating upstream backwater areas and downstream wake fields of substantial length (i.e. Young, 1991; Bennett et al., 2015). Such features were also observed during field experiments (Wenzel et al., 2014). This means that LW affects flow upstream and downstream in an area which is larger than the wood piece itself, which can be one reason for the slightly better simulation results in V1 compared to V2.

For both simulation variants, subsequent adjustment of riparian roughness coefficients is necessary to improve the goodness-of-fit. Only increasing riparian roughness by decreasing Strickler coefficients results in a smooth crest as it can be originally observed in the field experiments. As the calibrated roughness coefficients from the simulation without large wood are the baseline roughness for the simulations with wood, the riparian-zone roughness coefficients are calibrated to the flood extent of the conditions without large wood. Due to generally higher water levels in the field experiments and in the simulations with large wood, more water flows through a larger riparian area covered with vegetation. In the model, water flows too fast through adjacent riparian areas without subsequent adjustment of roughness. Emerged rigid elements such as riparian vegetation can lead to an increase of Manning's n and hence, a decrease of Strickler coefficients due to increasing friction exerted on flow (Shields et al., 2017). Therefore, a larger wetted area with generally low flow depths, a largely continuous cover of dense grassy vegetation as well as an uneven microtopography due to i.e. elevated grass root wads observed in adjacent riparian areas during field experiments could have led to the necessity of increasing local roughness in this study; especially due to the lack of such features in the model's calculation mesh.

Decreasing Strickler coefficients by 30 % in variant V1 and 55 % in LW affected sections only (V2) are in the range of previous studies. For instance, Gregory et al. (1985) detected an LW related increase in Manning's n by 48.5 % and Dudley et al. (1998)

show an average increase of 36 %. Furthermore, MacFarlane and Wohl (2003) compare streams with and without LW and find Darcy-Weisbach's f on average 58 % higher in streams containing in-channel LW. However, it should be noted that boundary conditions, such as discharge, river size, LW volume, etc. as well as the methodological approaches greatly vary between studies. For example, MacFarlane and Wohl (2003) investigate high-gradient mountain streams while Shields and Gippel (1995) focus on lowland rivers. This illustrates the need of a common framework for better comparability of studies on large wood previously proposed by Wohl et al. (2010). This becomes especially true regarding the influence of stable in-channel LW on roughness coefficients.

The results presented may only be valid for small, single-thread and steep rivers with a defined amount of stable large wood elements indicating the narrow boundary conditions of this study. When modelling the potential impact of stable large wood as a change of in-channel roughness coefficients with different boundary conditions and without data of large wood-influenced discharge for calibration, the application of ensemble-simulations with literature-based values of large wood induced increase of roughness may be used for a first assessment. Here, estimation methods for large wood induced roughness increase in small, high-gradient streams and rivers as previously developed by Shields and Gippel (1995) for large lowland rivers would be useful. Additionally, reviews of recent advances in research on the hydraulics of LW in fluvial systems would be highly beneficial, similar to recent reviews and meta-analyses addressing ecological implications (i.e. Roni et al., 2015), large wood dynamics (i.e. Ruiz-Villanueva et al., 2016a; Kramer and Wohl, 2017), related risks for anthropogenic infrastructure (i.e. De Cicco et al., 2018) and large wood in fluvial systems in general (Wohl, 2017).

## 5.3 Representation of in-channel LW as discrete elements

Simulation variant V3 generates the best simulated hydrograph in regard to its overall shape compared to the mean observed hydrograph of field experiments with LW indicating the best simulation of flow processes in the study reach. Therefore, the time-consuming incorporation of discrete elements is an appropriate starting point for an advancement of model implementation and further studies on the hydrodynamics of in-channel LW. However, variant V3 produces a temporal shift between mean simulated and observed flood hydrograph causing a slightly delayed rise and falling limb of the flood hydrograph and hence, a delayed passage of the flood wave at Thomson-weir 2. Natural discrete LW elements have a complex shape, which strongly varies from piece to piece (and over time) concerning their geometry with twigs, branches, needles and floating debris caught up in the twigs. This complex shape as well as a permeability of LW elements and jams cannot be implemented in depth-averaged hydrodynamic models in detail and has to be simplified. The simplified implementation in terms of element impermeability, dimensions and positions of wood pieces may result in too strong flow alterations, which in turn lead to higher amounts of water being retained in the study reach and thus, the temporal shift of the modelled hydrograph. Intense flow alterations may also account for the fact that a subsequent adjustment of riparian roughness coefficients is not required in variant V3, as too strong energy losses and flow declarations caused by discrete LW objects account for roughness originally caused by other roughness elements not represented in the calculation mesh such as riparian vegetation and microtopography.

Nevertheless, variant V3 still shows a very high goodness-of-fit. A similarly high Nash-Sutcliffe-Efficiency was obtained in the study of Keys et al. (2018), who use discrete weirs to represent large wood objects for simulating their effects on floodplain connectivity. However, although variant V3 reveals the best simulation result, the temporal shift results in a lower goodness-of-fit and hence, model quality compared to simulation variants V1 and V2. Therefore, solely relying on statistical goodness-of-fit indicators on such high spatio-temporal scale may not be sufficient and visual interpretation should not be excluded when assessing model results.

Although the roughness coefficient approach presented in this study is feasible with all models which are based on the SWE, only models enabling the simulation of two- and three-dimensional flow conditions can be used for the incorporation of simplified discrete large wood elements. In this study, only a single design of discrete large wood elements was incorporated as topographic features into the calculation mesh. Other designs may be also suitable such as discrete weirs (Keys et al., 2018) or arrays of pillars allowing water to flow through. Further research including a comparison of different designs of discrete large wood elements in 2D-simulations under equal boundary conditions could be beneficial. Furthermore, in the present study calibration is solely conducted using the hydrograph at Thomson-weir 2. As point measurements of flow depth, velocity and inundation extent in the field could improve model accuracy assessments, multi-criteria calibration approaches may be considered in future studies simulating the hydraulic effects of stable in-channel large wood.

## 6 Conclusion

The hydrodynamic simulations conducted in the present study show that average flood hydrographs of previously conducted field experiments without in-channel LW can be accurately simulated in the small and high-gradient study reach using HYDRO_AS-2D. Nevertheless, minor discrepancies need to be considered. The effect of stable in-channel LW was satisfactorily simulated using roughness coefficients. However, differences in model quality can be detected between increasing in-channel roughness in the entire reach or in LW affected channel sections only, where the latter results in a lower statistical goodness-of-fit. Visually, most accurate simulations of LW related impacts on flood hydrographs regarding its overall shape can be obtained using discrete large wood elements as proposed in previous studies (Smith et al., 2011) but comes with a temporal shift between observation and simulation due to the impermeability of the LW elements as well as a higher demand of effort and time required for their incorporation into the model (Table 3). Therefore, using channel roughness coefficients for simulating the impact of stable large wood elements on discharge time series suggests to be similarly accurate as the implementation of discrete elements on reach or larger (i.e. catchment) scale, where minor differences are smaller than the overall model uncertainty. Although constrained to the boundary conditions of this study, the simulation results indicate that the impact of stable in-channel large wood may be simulated with a reduced amount of time and work required for model set-up and incorporation of discrete large wood elements through the use of roughness coefficients. Thus, model-based impact assessments of, for instance, stream restoration measures considering stable large wood, may become more feasible; especially on larger scale or in less critical channel-sections, where a fully resolved flow assessment with three-dimensional models is

not required or practical. However, the present study is restricted to narrow boundary conditions, in turn illustrating the need for further research comparing methods of stable large wood incorporation in different models with varying model-dimensions and boundary conditions regarding channel morphology, large wood characteristics and water flow. Nevertheless, by comparing methods for simulating the impact of stable large wood on the reach scale, the present study can provide helpful

information for practical applications in modelling stable large wood related effects in small, first order streams and rivers.

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

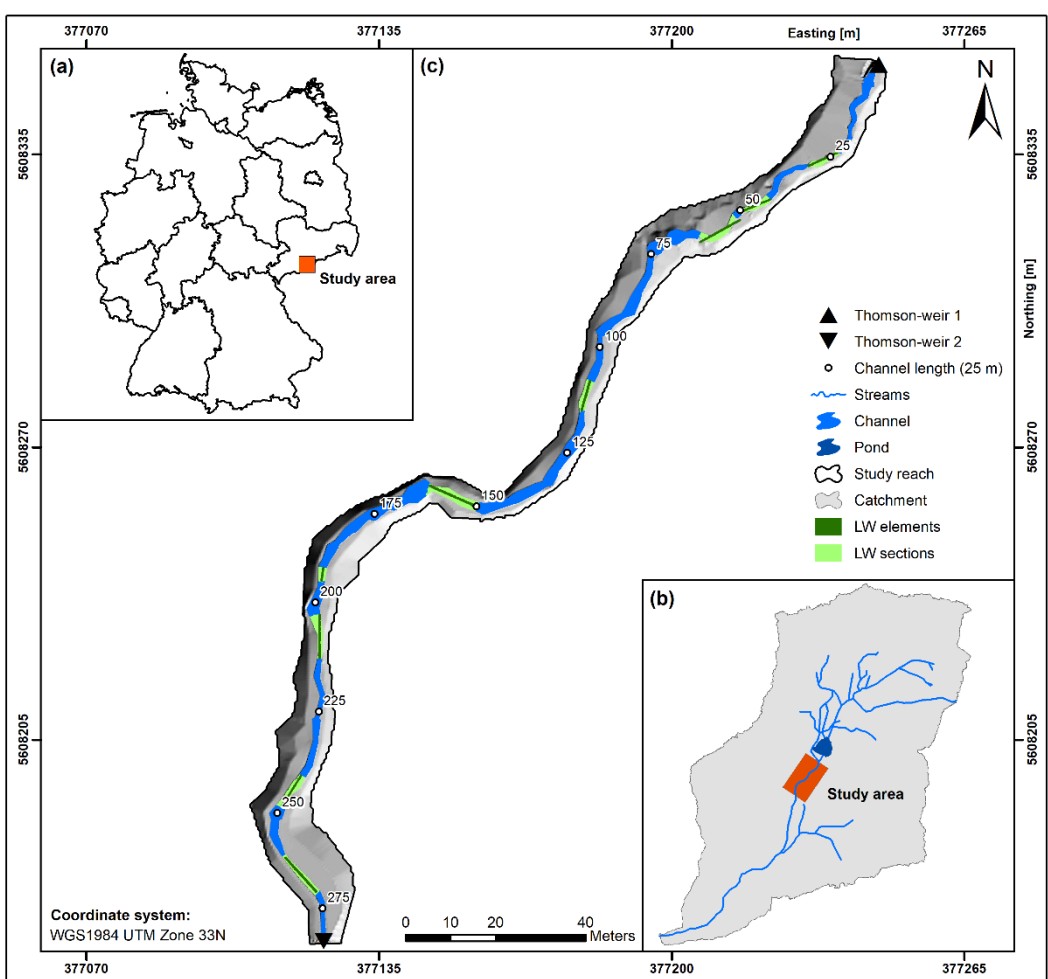

**Figure 1: (a) Location of the study area in Germany (administrative units: BKG, 2018) and (b) position of the study reach in the catchment of the Ullersdorfer Teichbächel (stream network: LVA, 2002). (c) LW affected sections and positions of discrete LW elements in the study reach.**

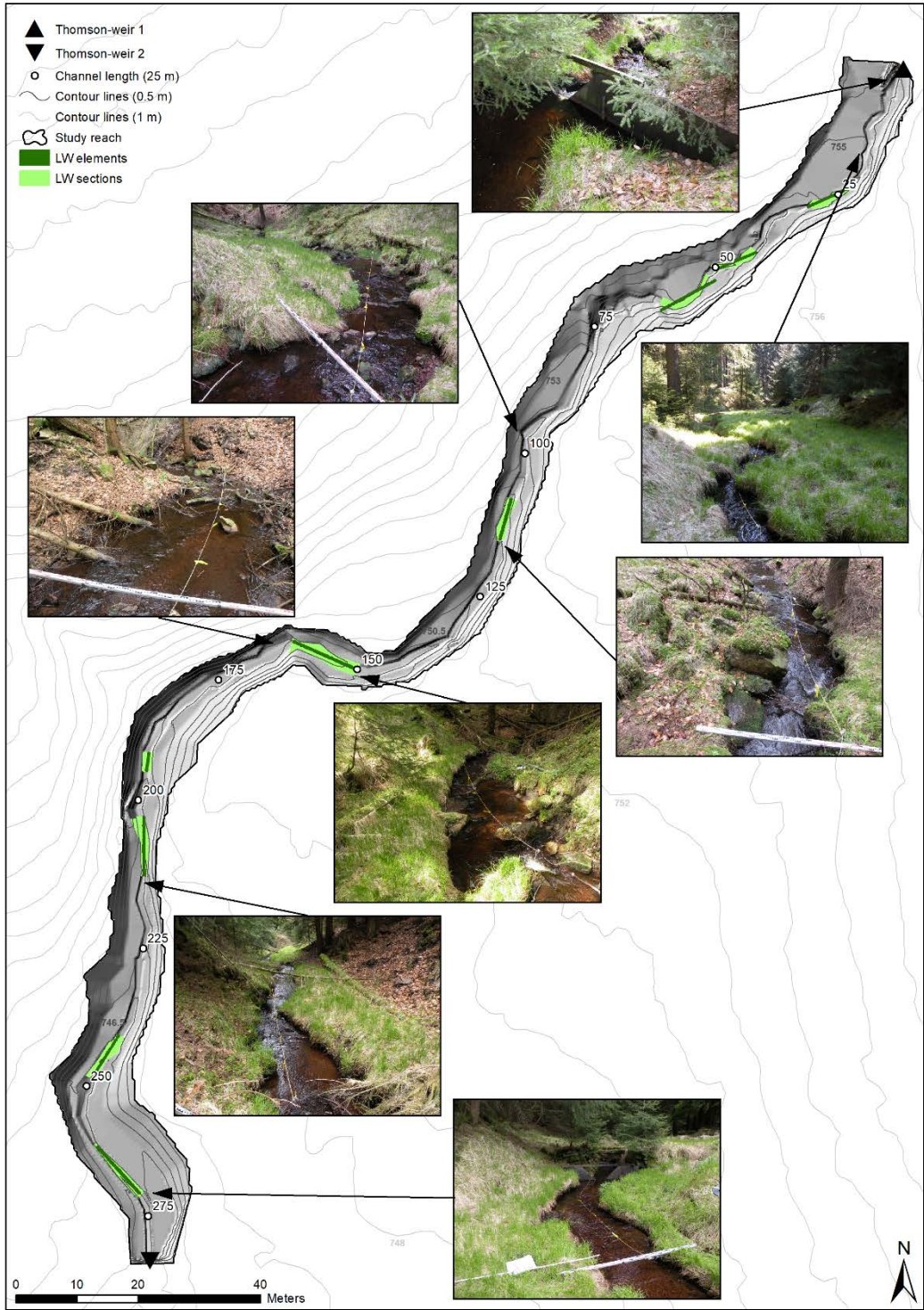

**Figure 2: Detailed map of the study reach (t**opographic data outside reach: GeoSN, 2008). Photographs were taken **9 years after the field experiments** in May 2017 in the direction of flow (north to south)**.**

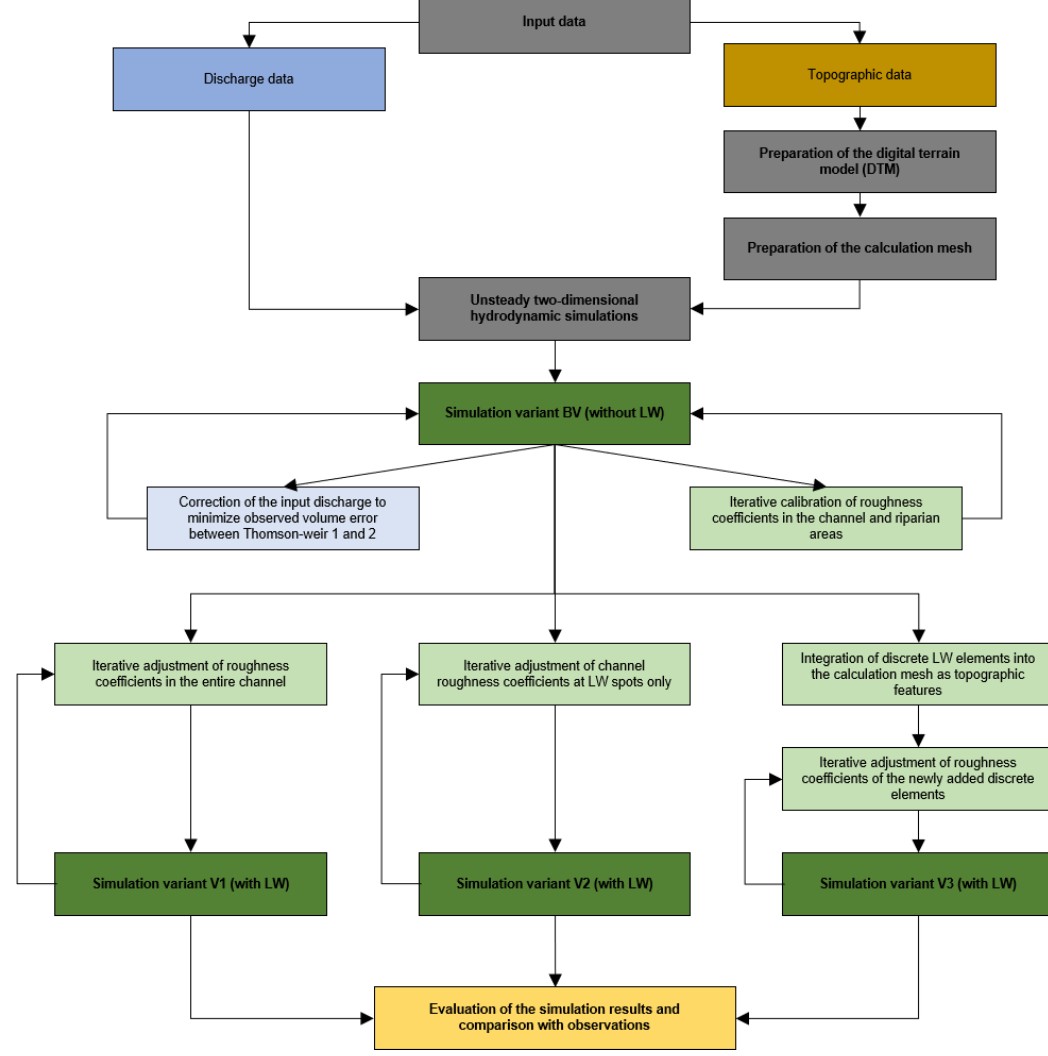

**Figure 3: Schematic illustration of the methodological workflow**

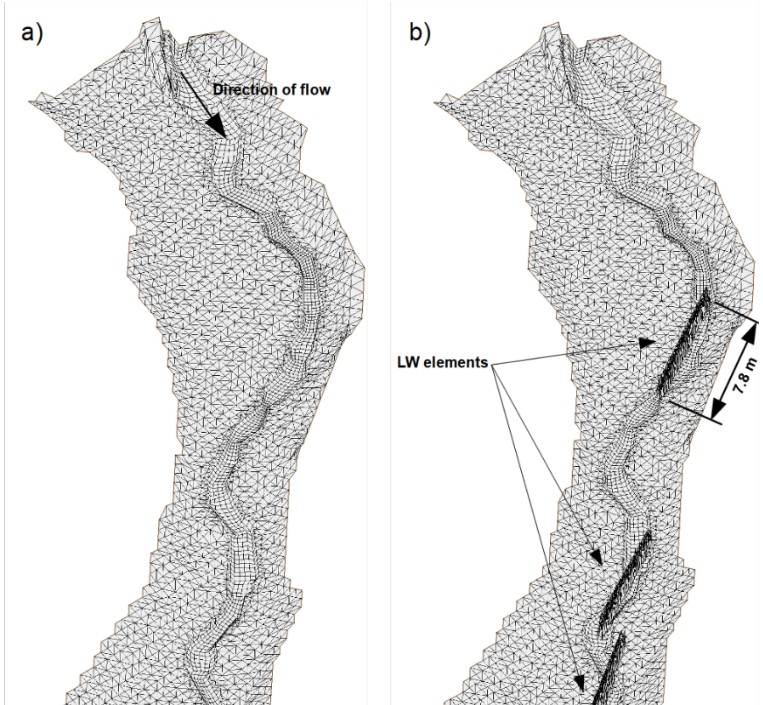

**Figure 4: a) Calculation mesh of the hydrodynamic model used in simulation variants RV, V1 and V2 with the use of variable Strickler coefficients adjusted for the entire channel (V1) or adjusted at the positions of all LW elements only (V2) and b) mesh with discrete LW elements used in variant V3. Example of the first 60 m of the study reach.**

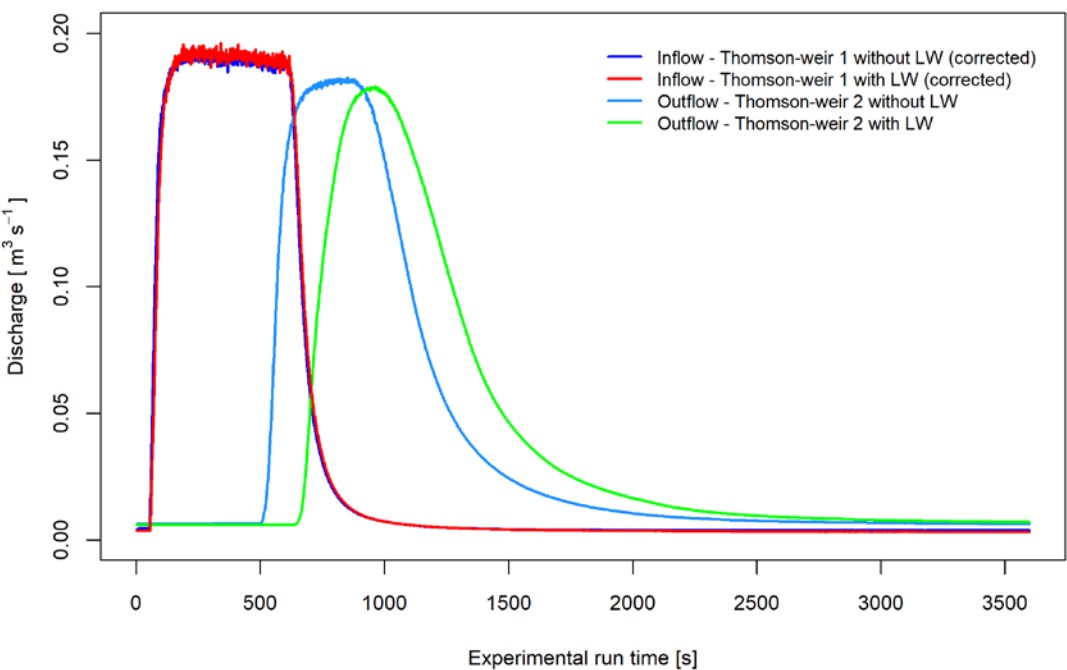

**Figure 5: Average measured and corrected flood hydrographs observed during field experiments with and without stable in-channel large wood (after Wenzel et al., 2014).**

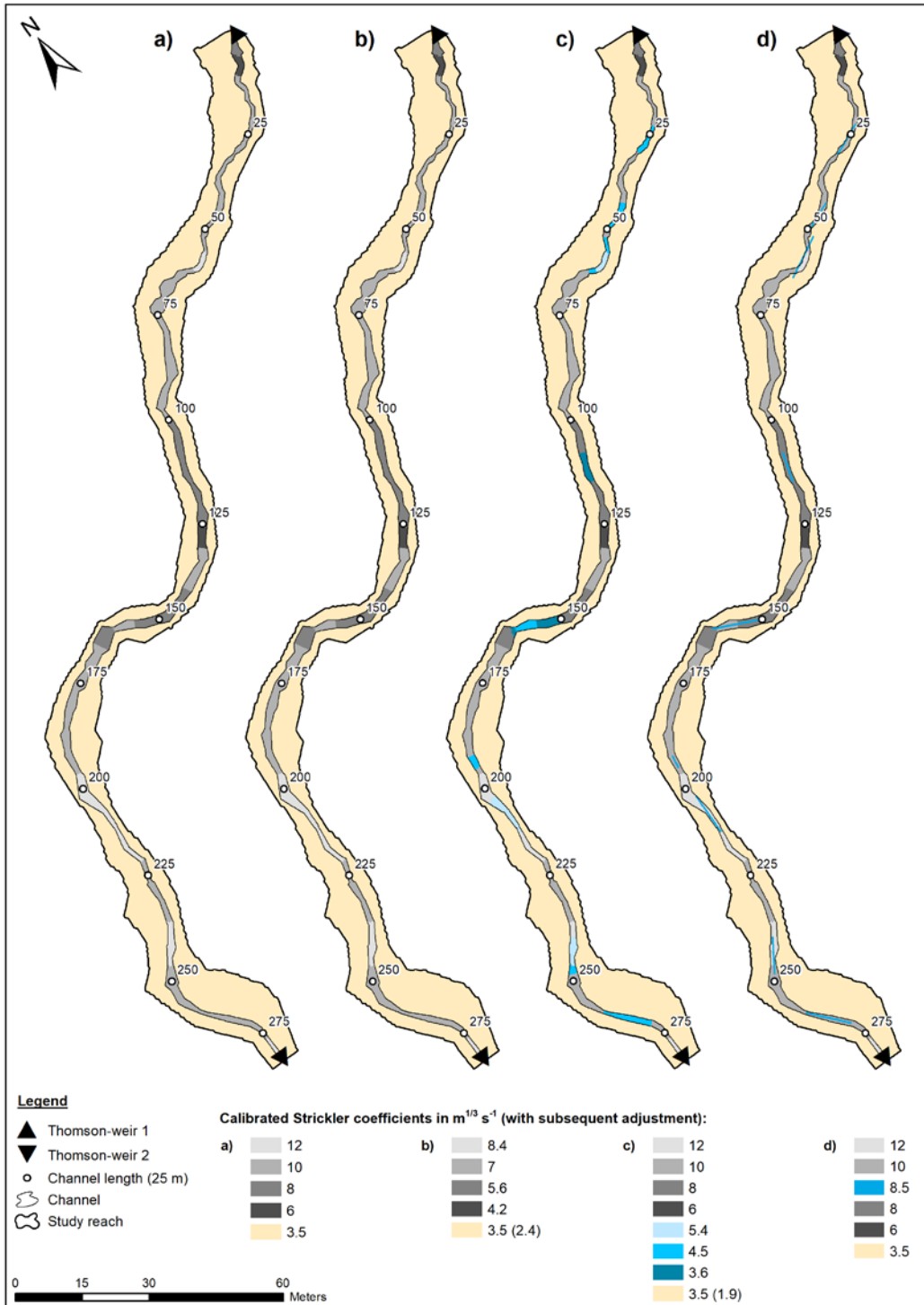

**Figure 6: Distribution of calibrated roughness coefficients of all simulation variants with and without LW in the study reach: a) reference variant RV without LW, b) variant V1 with stable LW as an increase of roughness in the entire channel, c) variant V2**

with stable LW as an increase of roughness at element positions only and, d) variant V3 with LW as discrete topographic elements of the calculation mesh. If displayed, values in brackets represent the Strickler coefficient after subsequent adjustment.

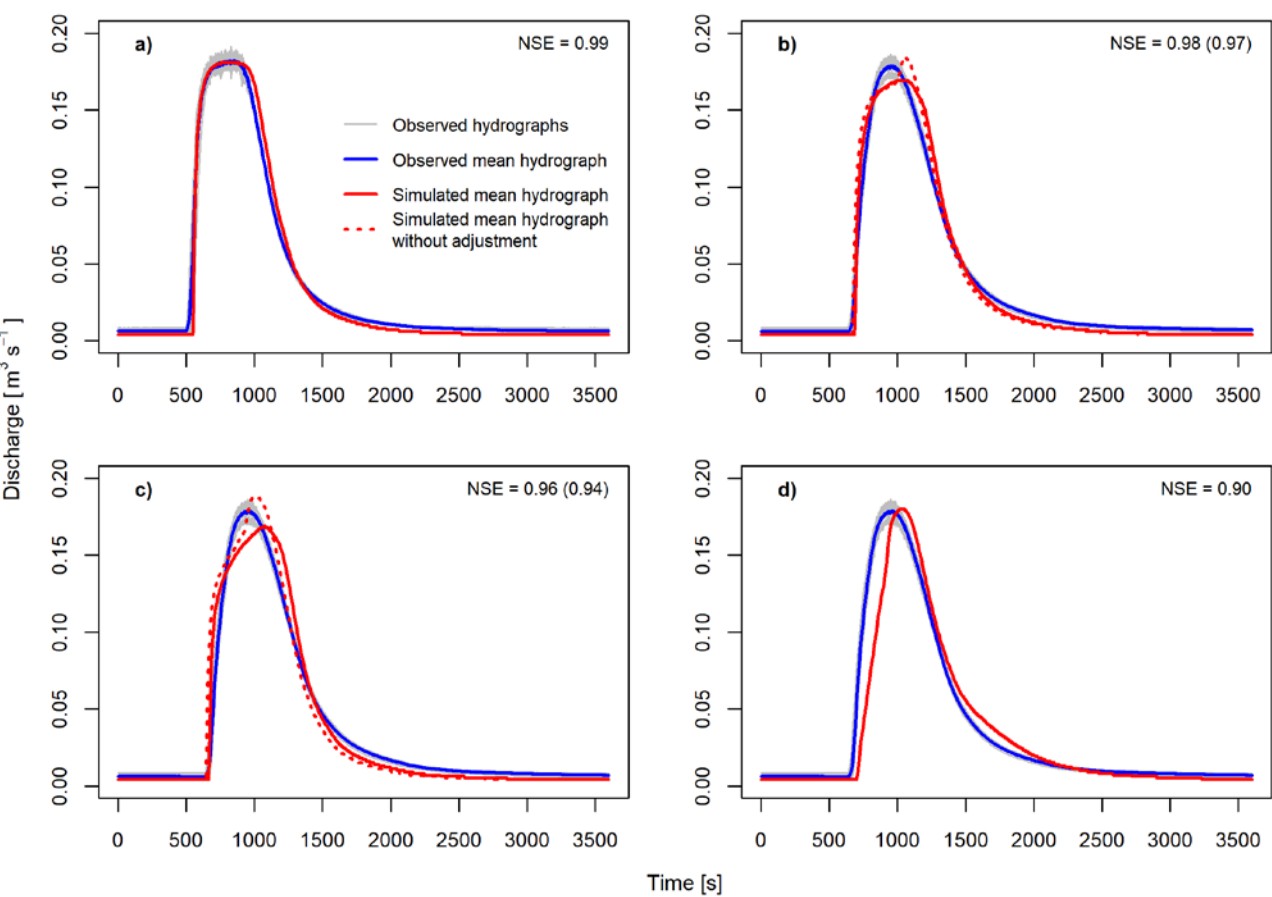

**Figure 7: Best simulated mean flood hydrographs of all simulation variants with and without LW at Thomson-weir 2: a) results of the reference variant RV without LW, b) variant V1 with stable LW as an increase of roughness in the entire channel, c) variant V2 with stable LW as an increase of roughness at element positions only and, d) variant V3 with LW as discrete topographic elements of the calculation mesh. For simulation variants V1 and V2 the best fit with and without subsequent adjustment of riparian Strickler coefficients is displayed. The Nash-Sutcliffe-Efficiency (NSE) is shown for each simulation variant. If displayed, values in brackets represent the NSE of simulations without adjustment of riparian roughness coefficients.**

**Table 1: Average observed and simulated discharge sums (m³ h⁻¹) at both Thomson-weirs for all simulation variants. For variant V1 and V2 discharge sums with subsequent adjustment of riparian Strickler coefficients are displayed.**

| Discharge sums (3600 s) for each variant (m³ h⁻¹) | Base-Variant | Variant 1 | Variant 2 | Variant 3 |
|---|---|---|---|---|
| Thomson-weir 1 (observed, corrected) | 128 | 128 | 128 | 128 |
| Thomson- weir 2 (observed) | 132 | 133 | 133 | 133 |
| Thomson-weir 1 (simulated) | 128 | 128 | 128 | 128 |
| Thomson-weir 2 (simulated) | 128 | 128 | 128 | 123 |
| Difference between observed and simulated values (Thomson-weir 2) | -4 | -5 | -5 | -10 |
| Observed difference between Thomson-weir 1 and 2 | -4 | -5 | -5 | -5 |

**Table 2: Calculated statistical goodness-of-fit parameters for all simulation variants. For variant V1 and V2 goodness-of-fit parameters with and without subsequent adjustment of riparian Strickler coefficients are displayed.**

| Goodness-of-fit parameters | Basie-Variant | Variant 1 without adjustment | Variant 1 | Variant 2 without adjustment | Variant 2 | Variant 3 |
|---|---|---|---|---|---|---|
| NSE | 0.99 | 0.97 | 0.98 | 0.94 | 0.96 | 0.90 |
| RSR | 0.11 | 0.18 | 0.14 | 0.24 | 0.19 | 0.32 |
| PBIAS (%) | - 3.5 | - 3.6 | - 3.7 | - 4.2 | - 4.0 | - 7.7 |

**Table 3: Attributes of approaches for large wood implementation applied in this study relative to the reference variant without large wood. Signs indicate an attribute being higher (+), lower (-) or equal (o) to the simulation without stable large wood.**

| Attribute | Variant V1 – reach-wise increase of roughness | Variant V2 – section-wise increase of roughness | Variant V3 – large wood as discrete elements |
|---|---|---|---|
| Work and time consumption | + | ++ | ++++ |
| Computational time | o | o | + |
| Statistical goodness-of-fit | - | -- | --- |
| Visual goodness-of-fit (hydrograph shape) | -- | -- | - |