# Peer review of "Hydrodynamic simulation of the effects of stable in-channel large wood on the flood hydrographs of a low mountain range creek, Ore Mountains, Germany"

_Hydrology and Earth System Sciences, 2019_

## Referee Comment (RC1) · Anonymous Referee #1 · 8 Mar 2019

1) General comments and my recommendation:

The authors have performed satisfactory analyses and generated positive results; The paper is well written and the subject matter is adequate to the scope of the journal, although I recommend this paper for publication after some structural revisions.

2) Minor corrections and observations:

a) Introduction:

Page 3, section 1, from line 30: I suggest to include also "Montgomery, D.R., Collins,

[Figure]

B.D., Buffington, J.M., Abbe, T.B., 2003. Geomorphic effects of wood in rivers. In: Gregory, S.V., Boyer, K.L., Gurnell, A.M. (Eds.), The Ecology and Management of Wood in World Rivers Bethesda (MD) American Fisheries Society, pp. 21–47.".

Page 4, section 1: I should advice the authors to demonstrate and stress why this paper is very important.

b) Study area:

I suggest to the authors to change the title of the chapter with "geological setting" including only the information about the catchment. The information about the study reach should be move into a new sub-chapter (e.g. study site) in "material and methods" chapter.

Page 4, section 2, between lines 15-25: If possible, the authors should provide the grain size distribution of the study reach.

Page 4, section 2, line 13: Please, you should add the longitude and latitude of the catchment.

c) Methods:

I suggest to change the title of the chapter with "material and methods".

Page 5, section 3.1, between lines 15-20: Please, remove the word "accurately" or give any quantity information about the accuracy.

Page 5, section 3.1, line 5: "...management application". Please, you should add a reference about it.

Page 5, section 3.1, line 17: Please, remove or give more information about the term "accurately".

Page 6, Section 3.2, between lines 30-5: Please, could you provide some information about the orientation of the LW placed in-channel? Were they placed cross-stream or

stream-wise?

Page 6, section 3.2, between lines 15-20: Please, provide the type of interpolation you used.

Page 8, section 3.4, lines 5 and 6: Please, change the unit of measure from cm to m. The authors should standardize the entire manuscript.

d) Results:

Page 8, section 4.1, line 20: "very well" is not a scientific statement.

Page 10, section 4.4, line 5: "very good" is not a scientific statement.

e) Discussions:

Page 10, section 5.1, lines 11 and 12: "very well" and "well" are not scientific statements.

Page 11, section 5.2, line 14: "well" is not a scientific statement.

Page 11, section 5.2, between lines 20-25: Please provide a reference in the literature about the sentence.

Page 12, section 5.3, line 23: "very good" is not a scientific statement.

Page 12, section 5.3, between lines 20-25: "...This is in accordance with previous studies...". Please, provide a reference about it.

f) Conclusions:

Page 13, section 6, between lines 18-20: "The effect of stable in-channel..." please, provide a reference about it.

In this chapter the authors are not properly writing the conclusions of the study conducted. Several parts should be moved to a new subchapter in the discussions part. For example, on page 13, section 6, between lines 10 and 15, the authors talk about a

limitation of the study. The same between lines 15 and 20. I think that you could talk about it in a new subchapter (e.g. limitations and future challenges), highlighting also the future development of the technique. Overall, in the conclusion the authors should present a concise and clear message, avoiding generalizations of the implications.

3) Suggestions:

a) As the author can read in the preface of the book of the First International conference of Wood in World Rivers (Gregory, S. V., K. L. Boyer, and A. M. Gurnell, editors. 2003. The ecology and management of wood in world rivers. American Fisheries Society, Symposium 37, Bethesda, Maryland), the term "debris" was first used to refer to the wood slash and debris left on the land and in the stream after timber harvest. For this reason, the term negatively connotes garbage or trash to the general public. The debate was reported also during the Third International conference of Wood in World Rivers in Padua (Italy) where the audience positive accepted to discourage further the use of the term "debris", encouraging the use of the word "wood". Thus, I would like to suggest to the authors to remove the term "debris" along the entire manuscript.

4) Figures:

Figure 1. I suggest to change the DTM of the study reach with another one that can give more information about the nature of the reach. An aerial photo could be enough.

Figure 4: I suggest to remove the titles and add the letters A, B, C, and D. Please, you should modify the text accordingly.

---

## Referee Comment (RC2) · Anonymous Referee #2 · 15 Mar 2019

1) General comments and my recommendation: After carefully reading the manuscript titled "Hydrodynamic simulation of the effects of in-channel large woody debris on the flood hydrographs of a low mountain range creek, Ore Mountains, Germany" and pondering the aspects of scientific relevance of the study and the specific findings and reflecting upon the coherence between the declared scope and practical utility of the work and the presented contents, I suggest that major revisions are necessary to enhance the manuscript and make it publishable in Hydrology and Earth System Sciences. The title of the presented work promises to simulate the effects of in-channel

LW on the flood hydrographs of a low mountain range creek and, therefore I expected insights on how the presence of LW affects the shape or form of the hydrograph and why. This would be of crucial importance for flood risk assessment. Instead the authors provided a detailed study on how to adjust the different model parameters (i.e. roughness both locally at single LW locations and globally on a reach scale) to obtain the best fit between measured hydrographs and simulated ones. I'm not contending that, per se, this exercise in not worth being done and hasn't been done rigorously and accurately; I rather surmise that the generated knowledge is only partially capable of explaining how hydromorphology linked to the presence of LW can be studied and the generated knowledge can, henceforth, inform decision makers in optimally implementing the water framework directive. Moreover, I miss a presentation and comparison of different hydrodynamic models capable to simulate different aspects of LW dynamics in rivers. The authors used HYDRO-AS-2D for their declared scope augmenting that this software is standardly employed in Germany (mainly for flood hazard assessment I suppose). This argumentation line is rather week. There should be a rigorous assessment of the best tool to be applied to analyze the considered processes. In the title I'd use the wording 'stable in-channel Large Wood', since, in essence, with the chosen modelling approach only stable LW can be considered, by adjusting the topographic mesh to the presence of these objects. It might well be the case that in the studied 282 m long section of the Ullersdorfer Teichbächl LW has been anchored to the river bed and morphodynamic change does not play a major role and, hence, given these circumstances HYDRO-AS-2D is applicable, but this mirrors only a minority of water courses in Europe. So, how can the generated knowledge be transferred to managers who have to deal with a broad variety of river systems? Imagine managers facing problems related to the WFD in very dynamic river system where LW is entrained and transported and interacts with obstacles continuously creating and destroying habitats, changing the river planform and its 3D structure, sorting sediment and, as a consequence of this conundrum of phenomena altering flood risk to a large extent. In such a case, working with HYDRO-AS-2D might not be the most recommendable option.

Given these considerations, I argue that attaching your work the broad scope of the WFD to enhance European rivers from various perspectives is a bit too far reaching and could inconveniently generate false expectations. In fact, you conclude the introduction by stating that "understanding its effects and the ability of predicting hydraulic impacts of LW in hydraulic simulations can be highly important for the use of LW in stream restoration projects and ecological-oriented management approaches in the scope of WLD", the paper, however, largely lacks a discussion on how, based on your findings, these ambitious goals can be accomplished. Based on the afore mentioned general comments I think that the introduction has to be reworked to assure full coherency between scopes, goals, what has been accomplished and how it contributes to the specific goals and the general scopes.

2) Specific comments:

Abstract: Personally the abstract is too long. As it is, I'd rather call it an extended abstract. I think that greater synthesis is required to inform the reader about tackled scientific problems, the adopted methodological approach (without details), a key message about the main finding and a brief concluding remark about the real broader implications of your work.

Section 1: Introduction: Page 2; Line 22: Instead of LWD, I'd use LW (Large Wood) which is the commonly accepted terms in the scientific community dedicated to wood in world rivers. Personally I think that the literature review about the LW hydrodynamic modelling is insufficient. There is much more out there that should be acknowledged and briefly described. Page 3; Line 13: The issue of flood risk due to LW mobility is introduced here. But how can one assess the LW contribution to flood risk if LW is assumed as fixed? Only through the change of local topography and roughness I suppose, which to my mind has to be better acknowledged as a limiting factor of the chosen approach.

Section 3: Methods: Beyond the above mentioned suggestion to compare existing

potentially applicable hydrodynamic models, I think that a figure with a workflow that explains the followed methodological steps might enhance the structure of the paper.

Section 4: Results: I've no particular objections to the obtained results.

Section 5: Discussion: To enhance this section, I invite the authors to carefully address the general concerns summarized in the first section of this review. Ideally departing from the obtained results one should be able to address the main issues which have been anticipated in the introduction either "positively" (i.e. underlining the contribution of the obtained results to the clarification of the risen issue) or "negatively" (i.e. expanding upon the necessity to integrate knowledge and to further investigate and specifically address open questions partially applicable findings).

3) Minor corrections and observations: With respect to the suggestion of minor corrections and observations I reaffirm the importance of carefully enhancing the paper according to amendments indicated by the anonymous referee 1.

―――――――――――――――――

---

## Author Comment (AC1) · 18 Apr 2019

On behalf of the authors we would like thank both anonymous reviewers for their constructive comments regarding our manuscript. We are certain that these comments greatly improve our manuscript and they will be incorporated in a revised version of the manuscript.

In the following section we will reply to all comments of both reviewers denoted with R1 (i.e. reviewer comment 1) and A1 (i.e. author response 1), respectively. As major parts of the manuscript including introduction, discussion, conclusion and abstract are modified based on the comments and suggestions of reviewer 1 and 2, we added the entire modified manuscript at the end of this response.

**Reviewer #1:**

**R1:** Page 3, section 1, from line 30: I suggest to include also "Montgomery, D.R., Collins, B.D., Buffington, J.M., Abbe, T.B., 2003. Geomorphic effects of wood in rivers. In: Gregory, S.V., Boyer, K.L., Gurnell, A.M. (Eds.), The Ecology and Management of Wood in World Rivers Bethesda (MD) American Fisheries Society, pp. 21–47.".

**A1:** We acknowledge the literature suggestion. After careful studying the reference suggested, it was added among others to the introduction chapter of the manuscript.

**R2:** Page 4, section 1: I should advice the authors to demonstrate and stress why this paper is very important.

**A2:** Based on the comment of anonymous reviewer 2, we modified large parts of the introduction, clarifying objectives and the relevance of our study.

**R3:** I suggest to the authors to change the title of the chapter with "geological setting" including only the information about the catchment. The information about the study reach should be move into a new sub-chapter (e.g. study site) in "material and methods" chapter.

**A3:** We disagree. In our opinion the description of the study reach should receive its own chapter and does not fit to the chapter "Methods" or "Material and methods". The name "Material and methods" suggests a description of the methods applied and materials used, which we find are the topographic datasets as well as the discharge data from field experiments but not the description of the nature of the study reach. In addition, we find that information about the geological setting of the entire catchment should and are already given in chapter 2 but should not receive its own chapter, because the description of the catchment geological setting is of minor relevance for the hydrodynamic simulation of the 282 m long study reach. However, to clarify the content of chapter 2, we changed the name from "Study area" to "Study reach".

**R4:** Page 4, section 2, between lines 15-25: If possible, the authors should provide the grain size distribution of the study reach.

**A4:** Detailed grain size analyses were not performed and thus, information about grain size distributions in the study reach cannot be provided.

**R5:** Page 4, section 2, line 13: Please, you should add the longitude and latitude of the catchment.

**A5:** We added the longitude and latitude of the catchment's centre (reference system WGS84) to the text.

**R6:** I suggest to change the title of the chapter with "material and methods".

**A6:** We changed the title of chapter 3 to "Material and methods".

**R7:** Page 5, section 3.1, between lines 15-20: Please, remove the word "accurately" or give any quantity information about the accuracy.

**A7:** We removed the word "accurately".

**R8:** Page 5, section 3.1, line 5: ": : :management application". Please, you should add a reference about it.

**A8:** We added additional references and modified the section in the following way:

Original:
"HYDRO_AS-2D was developed for practical applications in water management (Nujić, 2006) and is used in several studies simulating flow conditions in river sections (i.e. Lange et al. 2015) as well as in flood risk management applications.

Modification:
"HYDRO_AS-2D was developed for practical applications in water management (Nujić, 2006) and is used in several studies simulating flow conditions in river sections for flood risk management (i.e. Rieger and Disse, 2013) or with an ecological focus (i.e. Lange et al., 2015) and can produce a higher goodness-of-fit compared to other two-dimensional models as exemplarily shown in Lavoie and Mahdi (2017)."

**R9:** Page 5, section 3.1, line 17: Please, remove or give more information about the term "accurately".

**A9:** See response A7.

**R10:** Page 6, Section 3.2, between lines 30-5: Please, could you provide some information about the orientation of the LW placed in-channel? Were they placed cross-stream or stream-wise?

**A10:** In the field experiments (Wenzel et al., 2014), the large wood elements were placed lengthwise in the channel. This
information is already given in the original manuscript:

Original:

"The first 8 experimental runs were conducted with 9 large woody debris elements (spruce tree tops with a length ranging from 3 to 11.5 m, mean length 8.5 m), which were placed and fastened in the channel lengthwise 9 months earlier."

**R11:** Page 6, section 3.2, between lines 15-20: Please, provide the type of interpolation you used.

**A11:** We used the implementation of the procedure based on Hutchinson (1989) in the software environment ArcGIS v10.5
for interpolating the DTM. We added this information in the following way:

Original:

"The final DTM for the model is generated from processing and combining all topographic datasets in the software environment ArcGIS v10.5 (ESRI Inc., USA)."

Modification:

"The final DTM for the model is generated from processing and combining all topographic datasets in the software environment ArcGIS v10.5 (ESRI Inc., USA) using the implementation based on the procedure described in Hutchinson (1989) for interpolation."

**R12:** Page 8, section 3.4, lines 5 and 6: Please, change the unit of measure from cm to m. The authors should standardize the entire manuscript.

**A12:** We changed units from cm to m in the entire manuscript.

**R13:** Page 8, section 4.1, line 20: "very well" is not a scientific statement.

**A13:** We removed "very well" and modified the sentence:

Original:

"In general, the model simulates the characteristics of the observed hydrograph very well."

Modification:

"In general, the model closely simulates the characteristics of the observed hydrograph."

**R14**: Page 10, section 4.4, line 5: "very good" is not a scientific statement.

**A14:** We removed "very good" and modified the sentence accordingly:

Original:

"According to the classification of Moriasi et al. (2007), goodness-of-fit parameter values calculated for variant V3 as well as for all other simulation variants in this study indicate very good simulation results."

Modification:

"According to the classification of Moriasi et al. (2007), goodness-of-fit parameter values calculated for variant V3 as well as for all other simulation variants in this study indicate simulation results of high accuracy."

**R15:** Page 10, section 5.1, lines 11 and 12: "very well" and "well" are not scientific statements.

**A15:** We removed "very well" and "well" and modified the sentence:

Original:

"In general, the 2D hydrodynamic model mimics the flow conditions of the field experiments without LWD (variant BV) very well. Especially the time of rise, the rising limb and the flood peak are well represented, minor deviations can be observed along the hydrograph's falling limb only due to the broader shape of the simulated hydrograph."

Modification:
"In general, the 2D hydrodynamic model closely mimics the flow conditions of the field experiments without LW (variant BV). Especially the time of rise, the rising limb and the flood peak are accurately represented, minor deviations can be observed along the hydrograph's falling limb only due to the broader shape of the simulated hydrograph."

**R16:** Page 11, section 5.2, line 14: "well" is not a scientific statement.

**A16:** We removed "well" and modified the sentence:

Original:
"Compared to the simulation result of the mean observed hydrograph of the field experiments without in-channel LWD, variants V1 and V2 produce less well fitting simulated hydrographs, which is also indicated by the slightly lower values of statistical goodness-of-fit parameters."

Modification:
"Compared to the simulation result of the mean observed hydrograph of the field experiments without in-channel LW, variants V1 and V2 produce less closely fitting simulated hydrographs, which is also indicated by the slightly lower values of statistical goodness-of-fit parameters."

**R17:** Page 11, section 5.2, between lines 20-25: Please provide a reference in the literature about the sentence.

**A17:** This sentence depicts a conclusion drawn based on the previous sentence where the reference (Shields et al., 2017) is given. For clarification we conducted the following adjustments:

Original:
"Emerged riparian vegetation can lead to an increase of Manning's n and hence, a decrease of Strickler coefficients due to increasing friction exerted on flow (Shields et al., 2017). Therefore, generally low flow depths, a largely continuous cover of dense grassy vegetation as well as a uneven microtopography due to i.e. elevated grass root wads observed in adjacent riparian areas during field experiments may have led to the necessity of increasing local roughness; especially due to the lack of such features in the model's calculation mesh."

Modification:

"Emerged rigid elements such as riparian vegetation can lead to an increase of Manning's n and hence, a decrease of Strickler coefficients due to increasing friction exerted on flow (Shields et al., 2017). Therefore, generally low flow depths, a largely continuous cover of dense grassy vegetation as well as an uneven microtopography due to i.e. elevated grass root wads observed in adjacent riparian areas during field experiments could have led to the necessity of increasing local roughness in this study; especially due to the lack of such features in the model's calculation mesh."

**R18:** Page 12, section 5.3, line 23: "very good" is not a scientific statement.

**A18:** Based on our findings, in this sentence, we draw a first conclusion and evaluate the incorporation of simplified discrete elements based on our objective results. However, we modified the sentence in the following way:

Original:

"…indicating that discrete elements are a very good starting point for an advancement of model implementation and further studies on the hydrodynamics…"

Modification:

"…indicating that discrete elements are an appropriate starting point for an advancement of model implementation and further studies on the hydrodynamics…"

**R19:** Page 12, section 5.3, between lines 20-25: ": : :This is in accordance with previous studies: : :". Please, provide a reference about it.

**A19:** This sentence is an introductory statement referring to the references given in the following two sentences. To clarify this connection, we modified the original section in the following way:

Original:

"This is in accordance with previous studies using three-dimensional hydrodynamic models (computational fluid dynamics, CFD). On the one hand, general flow patterns caused by large wood can be simulated using impermeable discrete elements, when an accurate simulation of flow near LWD objects is neglectable (Xu and Liu, 2017). On the contrary, simplifications of

LWD objects made during the integration process into the calculation mesh may cause deviations and inaccuracies (Allen and Smith, 2012)."

Modification:

"This is in accordance with previous studies using three-dimensional hydrodynamic models (computational fluid dynamics, CFD): For example, on the one hand, general flow patterns caused by large wood can be simulated using impermeable discrete elements, when an accurate simulation of flow near LW objects is neglectable (Xu and Liu, 2017). On the contrary, simplifications of LW objects made during the integration process into the calculation mesh may cause deviations and
inaccuracies (Allen and Smith, 2012)."

**R20:** Page 13, section 6, between lines 18-20: "The effect of stable in-channel: : :" please, provide a reference about it.

**A20:** We nearly reformulated the entire chapter 6. The sentence was removed in that scope. See author response A21.

**R21:** In this chapter [conclusion] the authors are not properly writing the conclusions of the study conducted. Several parts should be moved to a new subchapter in the discussions part. For example, on page 13, section 6, between lines
10 and 15, the authors talk about a limitation of the study. The same between lines 15 and 20. I think that you could talk about it in a new subchapter (e.g. limitations and future challenges), highlighting also the future development of the technique. Overall, in the conclusion the authors should present a concise and clear message, avoiding generalizations of the implications.

**A21:** We agree with the reviewer, that limitations should rather be mentioned in the discussion chapter and that the conclusion
chapter should be formulated in a more precise way, focussing on the results obtained in this study. Therefore, we added a new sub-chapter (5.4 General limitations and implications for further research) to the discussion chapter and reformulated the conclusion chapter:

Original:

"**6. Conclusion**

[revised manuscript text omitted]

Modification:

"**5.4 General limitations and implications for further research**

The present case study investigates the impact of large wood on the flood hydrographs under stable (fastened) conditions. This is often done in model-based impact assessments (i.e. Hafs et al. 2014, Lange et al., 2015) but does not necessarily represent reality. Large wood stability depends on several hydrological and morphological factors (see Kramer and Wohl, 2017) and may mostly occur in small streams and rivers, where large wood elements are large compared to the channel dimensions (i.e. Gurnell et al., 2002). Consequently, the validity of the results presented is limited to these hydromorphological conditions. A first assessment of potential large wood transport and hence, mobility can be evaluated with the conceptual model presented in Kramer and Wohl (2017). If wood transport can be expected or wood elements are not fastened, i.e. in the scope of a restoration measure, hydrodynamic simulations of large wood dynamics may be necessary as presented in Ruiz-Villanueva et al. (2014).

In addition, the model results are restricted to the specific set-up of boundary conditions of the field experiments in Wenzel et al. (2014). Thus, the results are valid for i.e. the amount of large wood, its volume and orientation as well as the channel morphology and hydrological conditions of the field experiments but might not be transferable without adjustment. Further simulations of the approaches presented in this study with varying boundary conditions regarding channel morphology and discharge are necessary to validate the results and further compare approaches of incorporating stable large wood in hydrodynamic models. This is also true for the increase of roughness determined during calibration and resulting in the best fit of the model. When modelling the potential impact of stable large wood as a change of in-channel roughness coefficients with different boundary conditions and without data of large wood-influenced discharge for calibration, the application of ensemble-simulations with literature-based values of large wood induced increase of roughness may be used for a first assessment. Here, estimation methods for large wood induced roughness increase in small, high-gradient streams and rivers, as previously developed by Shields and Gippel (1995) for large lowland rivers or reviews of recent advances in research on the hydraulics of LW in fluvial systems would be highly beneficial, as it is the case for recent reviews and meta-analyses addressing ecological implications (i.e. Roni et al., 2015), large wood dynamics (i.e. Ruiz-Villanueva et al., 2016a; Kramer and Wohl, 2017), related risks for anthropogenic infrastructure (i.e. De Cicco et al., 2018) and large wood in fluvial systems in general (Wohl, 2017).

[revised manuscript text omitted]

**R22:** As the author can read in the preface of the book of the First International conference of Wood in World Rivers (Gregory, S. V., K. L. Boyer, and A. M. Gurnell, editors. 2003. The ecology and management of wood in world rivers. American Fisheries Society, Symposium 37, Bethesda, Maryland), the term "debris" was first used to refer to the wood slash and debris left on the land and in the stream after timber harvest. For this reason, the term negatively connotes garbage or trash to the general public. The debate was reported also during the Third International conference of Wood in World Rivers in Padua (Italy) where the audience positive accepted to discourage further the use of the term "debris", encouraging the use of the word "wood". Thus, I would like to suggest to the authors to remove the term "debris" along the entire manuscript.

**A22:** We agree that the term "large wood" should be used instead of "large woody debris" due to the positive ecological functions of large wood in fluvial environments. We changed "large woody debris" and "LWD" to "large wood" and "LW", respectively, in the entire manuscript, all figures and the title.

**R23:** Figure 1. I suggest to change the DTM of the study reach with another one that can give more information about the nature of the reach. An aerial photo could be enough.

**A23:** An aerial image does not prove helpful in the study reach because of the dense canopy cover in the catchment of the Ullersdorfer Teichbächel. Instead, we added a more detailed map of the study reach including contour lines of the study reach and surrounding areas. Furthermore, photographs taken in May 2017 are included, giving a better overview of the nature of the reach:

Additional figure:

[Figure]

**Figure 2: Detailed map of the study reach (topographic data outside reach: GeoSN, 2008). Photographs were taken in May 2017 in the direction of flow (north to south).**

**R24:** Figure 4: I suggest to remove the titles and add the letters A, B, C, and D. Please, you should modify the text accordingly.

**A24:** We removed the titles of sub-plots and modified the figure description according to the letters.

---

## Author Comment (AC2) · 18 Apr 2019

**R25:** General comments and my recommendation: After carefully reading the manuscript titled "Hydrodynamic simulation of the effects of in-channel large woody debris on the flood hydrographs of a low mountain range creek, Ore Mountains, Germany" and pondering the aspects of scientific relevance of the study and the specific findings and reflecting upon the coherence between the declared scope and practical utility of the work and the presented contents, I suggest that major revisions are necessary to enhance the manuscript and make it publishable in Hydrology and Earth System Sciences. The title of the presented work promises to simulate the effects of in-channel LW on the flood hydrographs of a low mountain range creek and, therefore I expected insights on how the presence of LW affects the shape or form of the hydrograph and why. This would be of crucial importance for flood risk assessment. Instead the authors provided a detailed study on how to adjust the different model parameters (i.e. roughness both locally at single LW locations and globally on a reach scale) to obtain the best fit between measured hydrographs and simulated ones. I'm not contending that, per se, this exercise in not worth being done and hasn't been done rigorously and accurately; I rather surmise that the generated knowledge is only partially capable of explaining how hydromorphology linked to the presence of LW can be studied and the generated knowledge can, henceforth, inform decision makers in optimally implementing the water framework directive. Moreover, I miss a presentation and comparison of different hydrodynamic models capable to simulate different aspects of LW dynamics in rivers. The authors used HYDRO-AS-2D for their declared scope augmenting that this software is standardly employed in Germany (mainly for flood hazard assessment I suppose). This argumentation line is rather week. There should be a rigorous assessment of the best tool to be applied to analyze the considered processes. In the title I'd use the wording 'stable in-channel Large Wood', since, in essence, with the chosen modelling approach only stable LW can be considered, by adjusting the topographic mesh to the presence of these objects. It might well be the case that in the studied 282 m long section of the Ullersdorfer Teichbächl LW has been anchored to the river bed and morphodynamic change does not play a major role and, hence, given these circumstances HYDRO-AS-2D is applicable, but this mirrors only a minority of water courses in Europe. So, how can the generated knowledge be transferred to managers who have to deal with a broad variety of river systems? Imagine managers facing problems related to the WFD in very dynamic river system where LW is entrained and transported and interacts with obstacles continuously creating and destroying habitats, changing the river planform and its 3D structure, sorting sediment and, as a consequence of this conundrum of phenomena altering flood risk to a large extent. In such a case, working with HYDRO-AS-2D might not be the most recommendable option. Given these considerations, I argue that attaching your work the broad scope of the WFD to enhance European rivers from various perspectives is a bit too far reaching and could inconveniently generate false expectations. In fact, you conclude the introduction by stating that "understanding its effects and the ability of predicting hydraulic impacts of LW in hydraulic simulations can be highly important for the use of LW in stream restoration projects and ecological-oriented management approaches in the scope of WLD", the paper, however, largely lacks a discussion on how, based on your findings, these ambitious goals can be accomplished. Based on the afore mentioned

general comments I think that the introduction has to be reworked to assure full coherency between scopes, goals, what has been accomplished and how it contributes to the specific goals and the general scopes.

**A25:** Comment R25 contains several aspects we would like to respond to. It should be noted, that we modified several large sections of the original manuscript and not all modifications are stated in the author responses separately. Therefore, the complete modified manuscript can be found at the end of this document:

**R25a:** "The title of the presented work promises to simulate the effects of in-channel LW on the flood hydrographs of a low mountain range creek and, therefore I expected insights on how the presence of LW affects the shape or form of the hydrograph and why. This would be of crucial importance for flood risk assessment. Instead the authors provided a detailed study on how to adjust the different model parameters (i.e. roughness both locally at single LW locations and globally on a reach scale) to obtain the best fit between measured hydrographs and simulated ones."

**A25a:** We agree. Information about the influence of stable large wood on flood hydrographs should be mentioned and were added to the introduction chapter.

**R25b:** "I rather surmise that the generated knowledge is only partially capable of explaining how hydromorphology linked to the presence of LW can be studied and the generated knowledge can, henceforth, inform decision makers in optimally implementing the water framework directive."

**A25b:** We agree. We will reformulate the introduction of the manuscript to precisely state the objectives of this study and an additional chapter was added to the discussion chapter summarising the limitations of the present study. This includes a differentiation between large wood that can be assumed as stable and those elements that a potentially mobile.

**R25c:** "Moreover, I miss a presentation and comparison of different hydrodynamic models capable to simulate different aspects of LW dynamics in rivers."

**A25c:** We partly agree. We will add further information on hydrodynamic modelling and modelling different aspects of large wood (stable and mobile) to the introduction chapter and give references on recent reviews containing information about hydrodynamic model applications and simulation large wood, but a full review of existing models and individual model capabilities is beyond the scope of this case study.

**R25d:** "The authors used HYDRO-AS-2D for their declared scope augmenting that this software is standardly employed in Germany (mainly for flood hazard assessment I suppose). This argumentation line is rather week. There should be a rigorous assessment of the best tool to be applied to analyze the considered processes."

**A25d:** We disagree. We do not argue that we use the model because it is one of the standard modelling systems in Germany. We use this model because it is capable of simulation high-gradient streams with an irregular shape and a high variance in depth and width (Nujić, 2006). For clarification we changed the corresponding section of the original manuscript in the following way.

Original:

"In this study, the two-dimensional hydrodynamic model HYDRO_AS-2D (version 2.2) is used to simulate the flow in the study reach with and without LWD. HYDRO_AS-2D was developed for practical applications in water management (Nujić, 2006) and is used in several studies simulating flow conditions in river sections (i.e. Lange et al. 2015) as well as in flood risk management applications. Especially in southern Germany and Austria, HYDRO_AS-2D became a standard 2D modelling system for hydrodynamic model applications (Faber et al., 2012). Due to the numerical approaches used in the modelling system, HYDRO_AS-2D is capable of accurately simulating mass exchange between channel and forelands, streams comprising hydraulic jumps, steep channel sections as well as high variability of channel width (Nujić, 2006)."

Modification:

"In this study, the two-dimensional hydrodynamic model HYDRO_AS-2D (version 2.2) is used to simulate the flow in the study reach with and without LW. HYDRO_AS-2D was developed for practical applications in water management (Nujić, 2006) and is used in several studies simulating flow conditions in river sections for flood risk management (i.e. Rieger and Disse, 2013) or with an ecological focus (i.e. Lange et al., 2015) and can produce a higher goodness-of-fit compared to other two-dimensional models as exemplarily shown in Lavoie and Mahdi (2017). Especially in southern Germany and Austria, HYDRO_AS-2D became a standard 2D modelling system for hydrodynamic model applications (Faber et al., 2012). Due to the numerical approaches used in the modelling system, HYDRO_AS-2D is capable of simulating mass exchange between channel and forelands, streams comprising hydraulic jumps, steep channel sections and a high variability of channel width as well as dike breaches (Nujić, 2006). The latter is to some extent comparable with the rapid release of water initiated by opening the flap gate weir used in the field experiments (see chapter 3.2). For the above-named reasons, HYDRO_AS-2D was chosen for the present study."

**R25e:** "In the title I'd use the wording 'stable in-channel Large Wood', since, in essence, with the chosen modelling approach only stable LW can be considered, by adjusting the topographic mesh to the presence of these objects."

**A25e:** We agree. The word "stable" was added accordingly.

**R25f:** "It might well be the case that in the studied 282 m long section of the Ullersdorfer Teichbächl LW has been anchored to the river bed and morphodynamic change does not play a major role and, hence, given these

5  circumstances HYDRO-AS-2D is applicable, but this mirrors only a minority of water courses in Europe. So, how can the generated knowledge be transferred to managers who have to deal with a broad variety of river systems? Imagine managers facing problems related to the WFD in very dynamic river system where LW is entrained and transported and interacts with obstacles continuously creating and destroying habitats, changing the river planform and its 3D structure, sorting sediment and, as a consequence of this conundrum of phenomena altering flood risk to a large extent. In such a case, working with

10  HYDRO-AS-2D might not be the most recommendable option. Given these considerations, I argue that attaching your work the broad scope of the WFD to enhance European rivers from various perspectives is a bit too far reaching and could inconveniently generate false expectations. In fact, you conclude the introduction by stating that "understanding its effects and the ability of predicting hydraulic impacts of LW in hydraulic simulations can be highly important for the use of LW in stream restoration projects and ecological-oriented management approaches in the scope of WLD", the paper, however, largely

15  lacks a discussion on how, based on your findings, these ambitious goals can be accomplished. Based on the afore mentioned general comments I think that the introduction has to be reworked to assure full coherency between scopes, goals, what has been accomplished and how it contributes to the specific goals and the general scopes."

**A25f:** We agree that linking or case study to the broad scope of the WFD might be too far reaching. According to the suggestion

20  of reworking the introduction, we removed the WFD from the introduction, redefined or goals and reformulated the introduction accordingly. The original and modified version of the introduction can be found below:

Original:

[revised manuscript text omitted]

5    incorporating large wood through a rather quick change of channel roughness coefficients and as time-demanding simplified mesh elements can be provide beneficial information for future studies simulating stable large wood related effects on stream hydraulics and ecology. This is underlined by Grabowski et al. (2019) who identified remaining uncertainties for the use of large wood in river restoration and natural flood risk management in practice. Knowledge gaps remain for instance regarding the alteration of channel roughness and hydraulic impacts such as backwater effects for the identification of local risks

10   (Grabowski et al., 2019) which can be addressed with hydrodynamic models.

Although limited to smaller streams and rivers were large wood jams and elements can be assumed as stable or situations in which large wood elements are fastened, the present study can contribute to the ability of predicting hydraulic impacts of stable in-channel large wood within hydrodynamic simulations and can also provide beneficial practical information for conducting simulation-based impact assessments of stream restoration projects considering stable large wood by comparing different

15   methods of large wood integration.”

**R26:** Abstract: Personally the abstract is too long. As it is, I'd rather call it an extended abstract. I think that greater synthesis is required to inform the reader about tackled scientific problems, the adopted methodological approach (without details), a

20   key message about the main finding and a brief concluding remark about the real broader implications of your work.

**A26:** We reworked the abstract in the following way:

Original:

[revised manuscript text omitted]

**R27:** Section 1: Introduction: Page 2; Line 22: Instead of LWD, I'd use LW (Large Wood) which is the commonly accepted terms in the scientific community dedicated to wood in world rivers.

**A27:** Corrected (see A22).

**R28:** Personally I think that the literature review about the LW hydrodynamic modelling is insufficient. There is much more out there that should be acknowledged and briefly described.

**A28:** We partly agree and added information about hydrodynamic modelling in general. Extensive reviews about LW hydrodynamic modelling are given elsewhere (i.e. Ruiz-Villanueva et al., 2016a, Bertoldi and Ruiz-Villanueva, 2017) and are named in the modified introduction (see A25). However, reviewing all aspects of LW hydrodynamic modelling such as large wood dynamics is beyond the scope of this case study focussing on stable large wood.

**R29:** Page 3; Line 13: The issue of flood risk due to LW mobility is introduced here. But how can one assess the LW contribution to flood risk if LW is assumed as fixed? Only through the change of local topography and roughness I suppose, which to my mind has to be better acknowledged as a limiting factor of the chosen approach.

**A29:** This study focusses on large wood under stable conditions. We acknowledge that this needs to be clarified as a limiting factor and was added in the modified introduction and the discussion chapter (see A25 and A21).

**R30:** Section 3: Methods: Beyond the above mentioned suggestion to compare existing potentially applicable hydrodynamic models, I think that a figure with a workflow that explains the followed methodological steps might enhance the structure of the paper.

**A30:** We added a figure to manuscript showing the schematic methodological workflow:

Additional figure:

[Figure]

**Figure 3: Schematic illustration of the methodological workflow**

**R31:** Section 5: Discussion: To enhance this section, I invite the authors to carefully address the general concerns summarized in the first section of this review. Ideally departing from the obtained results one should be able to address the main issues which have been anticipated in the introduction either "positively" (i.e. underlining the contribution of the obtained results to the clarification of the risen issue) or "negatively" (i.e. expanding upon the necessity to integrate knowledge and to further investigate and specifically address open questions partially applicable findings).

**A31:** We added a chapter to the discussion describing limitations of this study and implications for further research. Additionally, we reworked the conclusion to match the aims stated in the introduction (see A21).

**R32:** 3) Minor corrections and observations: With respect to the suggestion of minor corrections and observations I reaffirm the importance of carefully enhancing the paper according to amendments indicated by the anonymous referee 1.

**A32:** Done.

**Complete reworked manuscript with all modifications shown (red figures were modified):**

[revised manuscript text omitted]

**5.4 General limitations and implications for further research**

The present case study investigates the impact of large wood on the flood hydrographs under stable (fastened) conditions. This is often done in model-based impact assessments (i.e. Hafs et al. 2014, Lange et al., 2015) but does not necessarily represent reality. Large wood stability depends on several hydrological and morphological factors (see Kramer and Wohl, 2017) and may mostly occur in small streams and rivers, where large wood elements are large compared to the channel dimensions (i.e. Gurnell et al., 2002). Consequently, the validity of the results presented is limited to these hydromorphological conditions. A first assessment of potential large wood transport and hence, mobility can be evaluated with the conceptual model presented in Kramer and Wohl (2017). If wood transport can be expected or wood elements are not fastened, i.e. in the scope of a restoration measure, hydrodynamic simulations of large wood dynamics may be necessary as presented in Ruiz-Villanueva et al. (2014).

In addition, the model results are restricted to the specific set-up of boundary conditions of the field experiments in Wenzel et al. (2014). Thus, the results are valid for i.e. the amount of large wood, its volume and orientation as well as the channel morphology and hydrological conditions of the field experiments but might not be transferable without adjustment. Further simulations of the approaches presented in this study with varying boundary conditions regarding channel morphology and discharge are necessary to validate the results and further compare approaches of incorporating stable large wood in hydrodynamic models. This is also true for the increase of roughness determined during calibration and resulting in the best fit of the model. When modelling the potential impact of stable large wood as a change of in-channel roughness coefficients with different boundary conditions and without data of large wood-influenced discharge for calibration, the application of ensemble-simulations with literature-based values of large wood induced increase of roughness may be used for a first assessment. Here, estimation methods for large wood induced roughness increase in small, high-gradient streams and rivers, as previously developed by Shields and Gippel (1995) for large lowland rivers or reviews of recent advances in research on the hydraulics of LW in fluvial systems would be highly beneficial, as it is the case for recent reviews and meta-analyses addressing ecological implications (i.e. Roni et al., 2015), large wood dynamics (i.e. Ruiz-Villanueva et al., 2016a; Kramer and Wohl, 2017), related risks for anthropogenic infrastructure (i.e. De Cicco et al., 2018) and large wood in fluvial systems in general (Wohl, 2017).

Although the roughness coefficient approach presented in this study is feasible with all models which are based on the SWE, only models enabling the simulation of two- and three-dimensional flow conditions can be used for the incorporation of

simplified discrete large wood elements. Here, further restrictions may apply corresponding to the model-specific discretion methods and hence, restrictions regarding the design of the underlying calculation mesh. Thus, different models available should be compared with similar boundary conditions. This also true for the design of discrete LW elements as part of the calculation mesh. 
[revised manuscript text omitted]

---

## Referee Comment (RC3) · Daniel Scott (Referee) · 31 May 2019

**My background relevant to this manuscript**: I have led and collaborated on multiple investigations of wood in rivers, including investigations of how logs interact with and store bed sediment, the controls on wood loads in mountainous river basins, and wood jam dynamics (e.g., mobility) in both natural and engineered settings. With regards to wood-influenced hydrology, I have relatively little experience. Similarly, I have very little experience with hydrodynamic modeling, other than through coursework and collaborations.

With the editor's permission, I commented on the most recent version of this manuscript, from the file "hess-2019-35-AC2-supplement.pdf". The line numbers I refer to should match those given in that version of the manuscript, as shown in the pdf, with tracked changes shown.

**Broad comments:**

This manuscript essentially compares three techniques for modeling large wood-induced roughness in a 2D hydraulic model, using hydrograph shape as a calibration metric. This is a useful endeavor, and could aid applied hydraulic modeling (e.g., I consult with hydraulic modelers who regularly must include wood structures in 2D hydraulic models when evaluating river restoration designs). However, the paper as presented makes it difficult to extract the key results of the study, or even discern exactly what the contribution is. I think there's a great, short paper here, obscured by a presentation that doesn't do it justice. The science seems perfectly fine (and I trust the other reviewers, who are hopefully more experienced with hydraulic modeling than I am, to fully evaluate this). To bring out this paper's potential, I suggest the authors cut it down substantially to more clearly and concisely present their main message (i.e., under what circumstances is it best to model wood as discrete roughness elements, patches of roughness, or just a modulator of reach-scale roughness). I made comments throughout that, if addressed, will make substantial progress towards this goal. However, the authors understand their main message better than I likely do, so I suspect further revision will be necessary to make this an easy-to-digest paper that could then be a good resource for hydraulic modelers tasked with dealing with wood.

The abstract does not adequately explain why the modeling and field observations were compared. I made a line comment about this, but in general, consider providing a clearer explanation of your objectives. The results and implications are adequately explained, and it's clear from the first two sentences why we should care about wood's hydraulic effects, but the reasoning behind your methods is lacking. I think there should be some sentence to the effect of "We compared multiple methods of 2D hydraulic modeling to account for stable LW-induced roughness to determine which method most accurately simulates field-observed hydrographs and provide guidance for future hydraulic modeling of stable large wood."

Due to the aforementioned presentation issues, I want to check with you that your main message is coming across properly. I came away with the following main message: "Modeling wood as discrete roughness elements is not much more effective than simply adjusting reach-scale roughness, and desired model performance (i.e., whether you want a model with a high goodness-of-fit or a shape that replicates real hydrographs) is the only differentiating factor between these two approaches. Thus, you might as well just modulate reach-scale roughness

instead of going through the arduous work of modeling large wood structures as discrete roughness elements." I'll admit that in writing that, I'm a bit uncertain if that's actually the conclusion of this paper, which suggests that you need to make your main message more clear throughout the paper. Does that main message agree with your desired main message?

Overall, I like what I see as the main message of this paper, and I think the authors have a solid bit of work that is adequately presented but could be improved substantially to make a bigger impact. By revising the presentation, the authors can make this paper an effective resource to help the applied and research communities better understand hydraulic modeling of large wood. Even though this case study has limited application, it's a good incremental advance. I recommend major revisions to address the presentation issues I have brought up.

Sincerely,

Dan Scott (scott93@uw.edu)

**Line comments**:

Comments are organized by page and line number.

1,13: It strikes me as imprecise to say that wood can improve hydraulic and hydromorphological characteristics. Wood changes those things, but may or may not improve them, depending on one's valuation, although I certainly don't dispute that wood can "act positively on a river's ecology"! Consider rephrasing this statement to be less subjective.

1,14: I'm really happy to see a shift from "large woody debris" to just "large wood"!

1,23: What about the implementations are you testing? Answering that in a few words here will help guide readers through the rest of the abstract.

1,27: The writing here is unclear at times, and somewhat wordy. For instance, "Methodically, in-channel roughness coefficients are changed iteratively for retrieving the best fit between mean simulated and observed flood hydrographs with and without LW at the downstream reach outlet" Could instead be "We iterate in-channel roughness coefficients to best fit the mean simulated and observed flood hydrographs with and without LW at the downstream reach outlet" This is considerably shorter and easier to read, in my opinion. This is a style thing, but consider going through the manuscript (at least the abstract) and tightening up the wording to eliminate redundancy and imprecise verbiage. There are also some grammatical errors, likely stemming from the track changes, to watch out for (e.g., on line 1,29, there is a comma after an "and" that is out of place; there is a word missing on line 2,18). I won't comment on this further, as I'd rather focus on the scientific content and leave this to the authors and copyeditors. However, I suggest reading through the manuscript and editing for grammar and syntax.

1,31: Do you mean between the observed hydrographs and the model results? The statement as written implies a good fit between individual field observations.

3,14: This statement is likely untrue. For a more nuanced discussion of piece and jam mobility, see Kramer and Wohl (2017, *Geomorphology,* DOI: 10.1016/j.geomorph.2016.08.026). It might be safe to say that pieces longer than channel width are more likely to be stable. Reading on, you seem to acknowledge this, so it would be good to eliminate this contradiction.

4,4 – 4,14: This paper is about the hydraulic effects of wood, not wood mobilization. While all of this is interesting, I don't see how it has any bearing on this paper's objectives. Consider keeping the explanation of model dimensions (always a helpful thing to remind people of), but scrapping the review of wood transport modeling. The topic this paper addresses is plenty interesting, and doesn't really need this extraneous addition of wood mobility ideas to distract readers.

5,8 – 5,20: This addition is good, and seems to better explain your objectives. However, it is currently difficult to read, and a few sentences don't really fit with the overall purpose of the paragraph (to explain why you did this study). For instance, the last two sentences of this paragraph just says "Grabowski et al. (2019) highlight wood alternations of channel roughness and hydraulics as a knowledge gap in identifying local wood-induced risks." That sentence could be better placed at the beginning of this paragraph (or close to it) to motivate this study, as opposed at the end.

5,25: Instead of the vague "integration", consider "roughness modeling" or something similar.

Figure 6: It would be nice to show quantitative metrics of goodness-of-fit on these plots, to help with visual interpretation. One of these is best, and it would be nice if readers could quickly get that from this figure. Something in the caption might also work, but I just notice a lot of white space on the figure, so I feel that you could include this in the plots themselves. I know this information is in Table 2, but summarizing it in this figure would make this presentation more impactful.

10,14: To help people who may be unfamiliar with these goodness-of-fit metrics, please briefly define then in terms of what values indicate high goodness-of-fit and what values indicate the opposite, either here (just before you present the values, or as you present them) or in the methods.

11,2: I'm not sure I understand the justification for altering the previously-calibrated riparian-zone roughness coefficients. Wouldn't it be more rigorous to not alter these after calibration? Or, could you provide a process-based reason for altering them? Reading on, I see that you give this justification in the discussion. Consider alluding to that here to prevent readers from thinking the same thing I did.

11,9: The titles/names you use for each variant should be very consistent throughout the manuscript. That is, if you want to use "spots", make that so everywhere you mention this variant. That consistency will really help readers keep track of your arguments. A summary table, like the one I suggest below, would also be helpful.

13,15-23: I don't understand why this riparian roughness coefficient adjustment is necessary with the variants with wood but not the variant without wood. Is this due to an increase in wetted area to cells that include more vegetation? This might just be my misunderstanding, but consider

clearing this up a bit to justify why you adjusted riparian roughness in the wood-included models, but not the baseline model without wood.

15,24: Is this redundant with your statement on line 15,11? Here is an example where the organization of paragraphs and ideas is not clear. Why contrast V1 and V2, then switch to discussing riparian roughness, then switch back to contrasting V1 and V2? A more logical flow (i.e., making sure that each new idea builds on the last, and relates to the paper's main message) could help shorten and clear up this presentation.

14,7-10: Is this entire paragraph necessary? This sort of thing is well-covered in the introduction, and doesn't seem to need repeating here.

14,11-19: In this paragraph, you lead with some ideas (that discrete elements are simplified in 2D models), then eventually get to a point (that this could cause the behavior seen in V3). Consider leading with the point, then explaining it. That can really help readers keep track of your arguments and get more from your presentation.

Section 5.3: All of these paragraphs begin with "nevertheless", which makes me think that that word might not be necessary here. This section in general is difficult to parse and would be a good candidate for revision. Consider exactly what your main message is here and try to cut out whatever doesn't relate to it. For instance, is the discussion on lines 14,24-33 necessary? Reading it, I don't see how you clearly connect those papers to your work.

15, 8-16: By this point, it's clear that your results only apply to stable large wood. I don't think it's necessary to go through this explanation of how to evaluate wood stability. For starters, it's doubtful that the relationships given in Kramer and Wohl (2017) could even enable robust stability analysis, and hazard-focused wood stability analysis is better covered by other publications. Second, this paper isn't about wood mobility. You could clearly state in a single sentence that your results apply to small, single-thread, steep rivers with stable wood elements, and get the necessary idea across, without going into this level of detail that might derail a reader's attention.

15,26-31: This sentence is very long, and I'm unsure what you're trying to say. Consider cutting this down a bit and making the message clearer. For instance, as what "is the case"?

15,32: Is "SWE" defined anywhere else in the manuscript? I can't find it.

Section 5.4: In my opinion, these sections rarely are read, and often present information that is either obvious to the people who will actually be doing future work, or unnecessary for the people who won't be doing that work. Consider your audience here. Is it really necessary to explain all the ways this study could be improved? I could see a short paragraph stating what your results apply to (see comment on lines 26-31 of this page) being useful, but this reads as being unnecessary. Consider either shortening this section down to a few sentences, or integrating this information throughout the paper (where readers are more likely to actually read it). I know this section is in response to another reviewer's comment, but I suspect that this doesn't fully satisfy their comment either. It would be much more effective for readers to get this

information throughout the paper, instead of the current presentation, which somewhat undermines the results.

Section 6: Consider giving these conclusions in the discussion (throughout it) as well. Readers may get through the discussion wondering what the point of the analyses are, and then will need to get through the limitations sections before making it to the main point of the manuscript. I also suggest you clear up these points using something like a summary table. For instance, it could look something like the following:

| Roughness method | Pros | Cons |
| --- | --- | --- |
| V1 (reach-scale roughness adjustment) | | |
| V2 (roughness increase near LW) | | |
| V3 (discrete LW roughness elements) | | |

Such a table could give readers the essential information and recommendations from this modeling, put in context by a succinct discussion comparing the three modeling techniques you tested.

---

## Author Comment (AC3) · 24 Jun 2019

On behalf of the authors we would like thank the third reviewer Daniel N. Scott for his additional profound and helpful comments regarding the revised version of the original manuscript. When incorporated, we are certain that his comments greatly improve our manuscript towards a finalized version.

In the following section we will reply to all comments of our third reviewer denoted with R1 (i.e. reviewer comment 1) and

A1 (i.e. author response 1), respectively. As this review concerns the revised manuscript based on the review of both anonymous reviewers (available at https://www.hydrol-earth-syst-sci-discuss.net/hess-2019-35/hess-2019-35-AC2-supplement.pdf), we separate this author response from the previous and may only recapitulate comments and responses of both anonymous reviewers if necessary.

As major parts of the revised manuscript including introduction, discussion, conclusion and abstract are altered, we added the entire modified manuscript at the end of this response.

**Reviewer #3 (Daniel N. Scott)**

**R1:** 1,13: It strikes me as imprecise to say that wood can improve hydraulic and hydromorphological characteristics. Wood changes those things, but may or may not improve them, depending on one's valuation, although I certainly don't dispute that wood can "act positively on a river's ecology"! Consider rephrasing this statement to be less subjective.

**A1:** We agree, that this statement needs to be less subjective and modified it in the following way:

Original:
"The presence of large wood (LW) in river channels can improve the hydromorphological and hydraulic characteristics of rivers and streams and therefore act positively on a river's ecology."

Modification:
"Large wood (LW) can alter the hydromorphological and hydraulic characteristics of rivers and streams and may act positively on a river's ecology by i.e. leading to an increased habitat availability."

**R2:** 1,14: I'm really happy to see a shift from "large woody debris" to just "large wood"!

**A2:** We are glad that this modification is a consent between all reviewers.

**R3:** 1,23: What about the implementations are you testing? Answering that in a few words here will help guide readers through the rest of the abstract.

**A3:** We added a statement to the abstract. The modification is in accordance with the reviewer's suggestion and shown below:

Original:
"In this study, a two-dimensional hydraulic model is set up for a mountain creek to simulate the hydraulic effects of stable LW and to test different methods of LW implementation."

Modification:
"However, the work- and time-consumption varies between approaches of incorporating large wood in hydrodynamic models. In this study, a two-dimensional hydraulic model is set up for a mountain creek to simulate the hydraulic effects of stable LW and to compare multiple methods to account for large wood induced roughness. LW is implemented by changing in-channel roughness coefficients and by adding topographic elements to the model in order to determine which method most accurately simulates observed hydrographs and to provide guidance for future hydrodynamic modelling of stable large wood with two-dimensional models."

**R4:** 1,27: The writing here is unclear at times, and somewhat wordy. For instance, "Methodically, in-channel roughness coefficients are changed iteratively for retrieving the best fit between mean simulated and observed flood hydrographs with and without LW at the downstream reach outlet" Could instead be "We iterate in-channel roughness coefficients to best fit the mean simulated and observed flood hydrographs with and without LW at the downstream reach outlet" This is considerably shorter and easier to read, in my opinion. This is a style thing, but consider going through the manuscript (at least the abstract) and tightening up the wording to eliminate redundancy and imprecise verbiage. There are also some grammatical errors, likely stemming from the track changes, to watch out for (e.g., on line 1,29, there is a comma after an "and" that is out of place; there is a word missing on line 2,18). I won't comment on this further, as I'd rather focus on the scientific content and leave this to the authors and copyeditors. However, I suggest reading through the manuscript and editing for grammar and syntax.

**A4:** We are glad about this comment. The comma was removed and we changed the phrase according to the reviewer's suggestion:

Original:
"Methodically, in-channel roughness coefficients are changed iteratively for retrieving the best fit between mean simulated and observed flood hydrographs with and without LW at the downstream reach outlet and simplified discrete elements representing LW were incorporated into the calculation mesh."

Modification:
"We iterate in-channel roughness coefficients to best fit the mean simulated and observed flood hydrographs with and without LW at the downstream reach outlet. As an alternative approach of modelling LW induced effects, we use simplified discrete topographic elements representing individual LW elements in the channel."

**R5:** 1,31: Do you mean between the observed hydrographs and the model results? The statement as written implies a good fit between individual field observations.

**A5:** We agree. The sentence was modified in the following way:

Original:
"In general, the model results reveal a high goodness-of-fit of between the observed flood hydrographs of the field experiments without and with stable in-channel large wood. The best fit of simulation and mean observed hydrograph with in-channel LW

can be obtained when increasing in-channel roughness through decreasing Strickler coefficients - in the entire reach instead of a reduction at LW positions only."

Modification:

"In general, the simulations reveal a high goodness-of-fit of between the observed flood hydrographs and the model results without and with stable in-channel large wood. The best fit of simulation and mean observed hydrograph with in-channel LW can be obtained when increasing in-channel roughness coefficients in the entire reach instead of an increase at LW positions only"

**R6:** 3,14: This statement is likely untrue. For a more nuanced discussion of piece and jam mobility, see Kramer and Wohl (2017, *Geomorphology,* DOI: 10.1016/j.geomorph.2016.08.026). It might be safe to say that pieces longer than channel width are more likely to be stable. Reading on, you seem to acknowledge this, so it would be good to eliminate this contradiction.

**A6:** We agree that mobile and stable large wood may not be to distinguished with such simple metrics. To avoid this section
to be misleading, we modified it in the following way:

Original:
"Here, potentially mobile large wood and stable large wood have to be distinguished. Large wood assemblages and elements may be assumed stable when the median element length exceeds channel width (i.e. Gurnell et al., 2002), likely to occur in
small first order streams and rivers, which in turn are the most abundant order of water courses on the planet (Downing et al., 2012). However, even in small but steep headwater streams, large wood may be transported during hydrogeomorphic events of high magnitude such as debris flows (Galia et al., 2018) or extreme floods. A conceptual model for a first estimate of large wood transport in water courses is given in Kramer and Wohl (2017) including hydrological as well as morphological variables. Further detailed information about large wood dynamics in river networks can be found in recent reviews of Ruiz-Villanueva
et al. (2016a) and Wohl (2017). Potentially mobile large wood may drifts during floods, elements jam at bridges or other infrastructure and cause increased water levels, damage or completely destroy anthropogenic goods and structures (Schmocker and Hager, 2011). On the contrary, stable large wood remains in place, reduces water conveyance (Wenzel et al., 2014) and leads to increased water levels upstream and in turn, increased risk of flooding and water logging in surrounding areas. For these reasons, LW is removed from European rivers and streams for more than a century (Wohl, 2015) also to ensure
navigability in larger rivers (Young, 1991)."

Modification:
"Large wood assemblages and elements are more likely to be stable when their length exceeds channel width (i.e. Gurnell et al., 2002), most likely to occur in small first order streams and rivers, which in turn are the most abundant order of water courses on the planet (Downing et al., 2012). However, even in small but steep headwater streams, large wood may be transported during hydrogeomorphic events of high magnitude such as debris flows (Galia et al., 2018) or extreme floods. A conceptual model for a first estimate of large wood transport in water courses is given in Kramer and Wohl (2017) including hydrological as well as morphological variables. Further detailed information about large wood dynamics in river networks can be found in recent reviews of Ruiz-Villanueva et al. (2016a) and Wohl (2017). Large wood may drift during floods, elements jam at bridges or other infrastructure and cause increased water levels, damage or completely destroy anthropogenic goods and structures (Schmocker and Hager, 2011). On the contrary, stable large wood reduces water conveyance (Wenzel et al., 2014) and leads to increased water levels upstream and in turn, increased risk of flooding and water logging in surrounding areas. For these reasons, LW is removed from European rivers and streams for more than a century (Wohl, 2015) also to ensure navigability in larger rivers (Young, 1991)."

**R7:** 4,4-4,17: This paper is about the hydraulic effects of wood, not wood mobilization. While all of this is interesting, I don't see how it has any bearing on this paper's objectives. Consider keeping the explanation of model dimensions (always a helpful thing to remind people of), but scrapping the review of wood transport modeling. The topic this paper addresses is plenty interesting, and doesn't really need this extraneous addition of wood mobility ideas to distract readers.

**A7:** Despite partly in contrast to comments of the anonymous reviewers, we agree that the focus of this paper should be on stable large wood and the simulation of its effects on water flow. However, to maintain a consent between all reviewers on the one hand and avoiding distraction of readers on the other, we keep the information that both, large wood transport and the hydraulic impact of stable large wood, can be simulated with hydrodynamic models but remove further information on LW transport modelling studies.

Original:

[revised manuscript text omitted]

**R10:** Figure 6: It would be nice to show quantitative metrics of goodness-of-fit on these plots, to help with visual interpretation. One of these is best, and it would be nice if readers could quickly get that from this figure. Something in the caption might also work, but I just notice a lot of white space on the figure, so I feel that you could include this in the plots themselves. I know this information is in Table 2, but summarizing it in this figure would make this presentation more impactful.

**A10:** We agree. However, we only added the NSE as a widely used metric to the plot in order to avoid extensive redundancies with table 2 and to prevent subplots from becoming unclear.

Original:

[revised manuscript text omitted]

**R13:** 11,9: The titles/names you use for each variant should be very consistent throughout the manuscript. That is, if you want to use "spots", make that so everywhere you mention this variant. That consistency will really help readers keep track of your arguments. A summary table, like the one I suggest below, would also be helpful.

**A13:** We agree and replaced the word "spots" with LWD "sections" throughout the manuscript.

**R14:** 13,15-23: I don't understand why this riparian roughness coefficient adjustment is necessary with the variants with wood but not the variant without wood. Is this due to an increase in wetted area to cells that include more vegetation? This might just be my misunderstanding, but consider clearing this up a bit to justify why you adjusted riparian roughness in the wood-
included models, but not the baseline model without wood.

**A14:** Yes, this is correct. The calibrated roughness coefficients from the simulation without wood are the baseline roughness for the simulations with wood. Thus, the riparian-zone roughness coefficients are calibrated to the flood extent (and hence, influence of riparian roughness elements such as vegetation) of the conditions without wood. However, in the field experiments
and in the simulations with large wood the water level is higher, resulting in generally more water flowing through a larger riparian area. As pointed out in R14, a larger wetted area covered with vegetation and hence, different flow conditions between the variants with and without wood in the riparian zone could have led to the necessity of adjusting riparian Strickler coefficients.
We added more detailed information to the paragraph.

Original:

"For both simulation variants, subsequent adjustment of riparian roughness coefficients is necessary to improve the goodness-of-fit. Only increasing riparian roughness by decreasing Strickler coefficients results in a smooth crest as it can be originally observed in the field experiments. In the model, water flows too fast through adjacent riparian areas without subsequent
adjustment of roughness. Emerged rigid elements such as riparian vegetation can lead to an increase of Manning's n and hence, a decrease of Strickler coefficients due to increasing friction exerted on flow (Shields et al., 2017). Therefore, generally low flow depths, a largely continuous cover of dense grassy vegetation as well as an uneven microtopography due to i.e. elevated grass root wads observed in adjacent riparian areas during field experiments could have led to the necessity of increasing local roughness in this study; especially due to the lack of such features in the model's calculation mesh."

Modification:

"For both simulation variants, subsequent adjustment of riparian roughness coefficients is necessary to improve the goodness-of-fit. Only increasing riparian roughness by decreasing Strickler coefficients results in a smooth crest as it can be originally observed in the field experiments. As the calibrated roughness coefficients from the simulation without large wood are the baseline roughness for the simulations with wood, the riparian-zone roughness coefficients are calibrated to the flood extent of the conditions without large wood. Due to generally higher water levels in the field experiments and in the simulations with large wood, more water flows through a larger riparian area covered with vegetation. In the model, water flows too fast through adjacent riparian areas without subsequent adjustment of roughness. Emerged rigid elements such as riparian vegetation can lead to an increase of Manning's n and hence, a decrease of Strickler coefficients due to increasing friction exerted on flow (Shields et al., 2017). Therefore, a larger wetted area with generally low flow depths, a largely continuous cover of dense grassy vegetation as well as an uneven microtopography due to i.e. elevated grass root wads observed in adjacent riparian areas during field experiments could have led to the necessity of increasing local roughness in this study; especially due to the lack of such features in the model's calculation mesh."

**R15:** 15,24: Is this redundant with your statement on line 15,11? Here is an example where the organization of paragraphs and ideas is not clear. Why contrast V1 and V2, then switch to discussing riparian roughness, then switch back to contrasting V1 and V2? A more logical flow (i.e., making sure that each new idea builds on the last, and relates to the paper's main message) could help shorten and clear up this presentation.

**A15:** Yes, this statement is rather redundant. We removed it and rearranged section 5.2:

Original:

[revised manuscript text omitted]

**R16-18:** 14,7-10: Is this entire paragraph necessary? This sort of thing is well-covered in the introduction, and doesn't seem to need repeating here.

14,11-19: In this paragraph, you lead with some ideas (that discrete elements are simplified in 2D models), then eventually get to a point (that this could cause the behavior seen in V3). Consider leading with the point, then explaining it. That can really help readers keep track of your arguments and get more from your presentation.

Section 5.3: All of these paragraphs begin with "nevertheless", which makes me think that that word might not be necessary here. This section in general is difficult to parse and would be a good candidate for revision. Consider exactly what your main message is here and try to cut out whatever doesn't relate to it. For instance, is the discussion on lines 14,24-33?
Reading it, I don't see how you clearly connect those papers to your work

**A16-18:** We agree that section 5.3 requires revision to make it easier to follow and that removal of unnecessary information is needed. We modified it in the following way:

Original:

[revised manuscript text omitted]

**R19-22:** 15, 8-16: By this point, it's clear that your results only apply to stable large wood. I don't think it's necessary to go through this explanation of how to evaluate wood stability. For starters, it's doubtful that the relationships given in Kramer and Wohl (2017) could even enable robust stability analysis, and hazard-focused wood stability analysis is better covered by other publications. Second, this paper isn't about wood mobility. You could clearly state in a single sentence that your results apply to small, single-thread, steep rivers with stable wood elements, and get the necessary idea across, without going into this level of detail that might derail a reader's attention.

15,26-31: This sentence is very long, and I'm unsure what you're trying to say. Consider cutting this down a bit and making the message clearer. For instance, as what "is the case"?

15,32: Is "SWE" defined anywhere else in the manuscript? I can't find it.

Section 5.4: In my opinion, these sections rarely are read, and often present information that is either obvious to the people who will actually be doing future work, or unnecessary for the people who won't be doing that work. Consider your audience here. Is it really necessary to explain all the ways this study could be improved? I could see a short paragraph stating what your results apply to (see comment on lines 26-31 of this page) being useful, but this reads as being unnecessary. Consider either shortening this section down to a few sentences, or integrating this information throughout the paper (where readers are more likely to actually read it). I know this section is in response to another reviewer's comment, but I suspect that this doesn't fully satisfy their comment either. It would be much more effective for readers to get this information throughout the paper, instead of the current presentation, which somewhat undermines the results.

**A19-22:** We agree to point 19-22 and completely removed section 5.4. We added important information from this section to the end of discussion sections 5.2 and 5.3. SWE was not defined yet, we defined it in the introduction chapter.

Added to section 5.2:

"The results presented may only be valid for small, single-thread and steep rivers with a defined amount of stable large wood elements indicating the narrow boundary conditions of this study. When modelling the potential impact of stable large wood as a change of in-channel roughness coefficients with different boundary conditions and without data of large wood-influenced discharge for calibration, the application of ensemble-simulations with literature-based values of large wood induced increase of roughness may be used for a first assessment. Here, estimation methods for large wood induced roughness increase in small, high-gradient streams and rivers as previously developed by Shields and Gippel (1995) for large lowland rivers would be useful. Additionally, reviews of recent advances in research on the hydraulics of LW in fluvial systems would be highly beneficial, similar to recent reviews and meta-analyses addressing ecological implications (i.e. Roni et al., 2015), large wood dynamics (i.e. Ruiz-Villanueva et al., 2016a; Kramer and Wohl, 2017), related risks for anthropogenic infrastructure (i.e. De Cicco et al., 2018) and large wood in fluvial systems in general (Wohl, 2017)."

Added to section 5.3:

"Although the roughness coefficient approach presented in this study is feasible with all models which are based on the SWE, only models enabling the simulation of two- and three-dimensional flow conditions can be used for the incorporation of simplified discrete large wood elements. In this study, only a single design of discrete large wood elements was incorporated as topographic features into the calculation mesh. Other designs may be also suitable such as discrete weirs (Keys et al., 2018)

or arrays of pillars allowing water to flow through. Further research including a comparison of different designs of discrete large wood elements in 2D-simulations under equal boundary conditions could be beneficial. Furthermore, in the present study calibration is solely conducted using the hydrograph at Thomson-weir 2. As point measurements of flow depth, velocity and inundation extent in the field could improve model accuracy assessments, multi-criteria calibration approaches may be considered in future studies simulating the hydraulic effects of stable in-channel large wood."

**R23:** Section 6: Consider giving these conclusions in the discussion (throughout it) as well. Readers may get through the discussion wondering what the point of the analyses are, and then will need to get through the limitations sections before making it to the main point of the manuscript. I also suggest you clear up these points using something like a summary table. For instance, it could look something like the following:

| Roughness method | Pros | Cons |
|---|---|---|
| V1 (reach-scale roughness adjustment) | | |
| V2 (roughness increase near LW) | | |
| V3 (discrete LW roughness elements) | | |

Such a table could give readers the essential information and recommendations from this modeling, put in context by a succinct discussion comparing the three modeling techniques you tested.

**A23:** We agree and added a concluding sentence to sections 5.2 and 5.3 to connect the results to the aims of our study. In addition, we slightly modified the conclusion and added a table summarizing the results and conclusions of our study in a relative way.

[revised manuscript text omitted]

**Complete reworked manuscript with all modifications shown (red figures were modified):**

---

## Referee Report (RR1)

[referee-annotated manuscript omitted]

---

## Author Response (AR2)

On behalf of the authors we would like thank both reviewers for their helpful comments regarding the revised version of the original manuscript. In the following section we will reply to all comments of both reviewers denoted with R1 (i.e. reviewer comment 1) and A1 (i.e. author response 1), respectively.

In addition, we removed typing and formatting mistakes not explicitly mentioned by the reviewers.

We added the entire modified manuscript at the end of this response.

[Figure]

**Reviewer #1**

**R1:** 1,14: consider using anchored instead

**A1:** We changed the word accordingly.

**R2:** 1,24: delete one fullstop here

10   **A2:** Full stop deleted.

**R3:** 1,30: consider using the term adjustment

**A3:** We changed the word accordingly.

**R4:** 4,26: consider using anchored

**A4:** We changed the word accordingly.

20   **R5:** 5,10: please search for a better way to express the highly varied bed morphology. In other words please simplify

**A5:** We simplified the phrase in the following way:

Original:

25   "Beside a highly variable stream width, alternating slope gradients and grain sizes lead to a highly diverse distribution of stream depth along the study reach and hence, a generally complex channel structure."

Modification:

"Beside a highly variable stream width, alternating slope gradients and grain sizes lead to an alternation of stream depth along

30   the study reach and hence, a generally complex channel structure."

**R6:** 5,15: prevailing?

**A6:** We changed the word accordingly.

**R7:** 5,21: I'd skip "in length" here.

**A7:** We agree and removed it.

**R8:** 8,10: consider substituting with reference variant RV

**A8:** We agree, that reference variant RV seems more suitable because its calibrated roughness coefficients are used as the baseline reference for the variants 1 to 3. We replaced base variant BV to reference variant RV in the entire manuscript.

**R9:** 9,9: see previous comment

**A9:** See author response A8.

15 **R10:** 12,27-29: eliminate the blank ine between the paragraphs

**A10:** Done.

**R11:** 14,15: Eliminate this unnecessary fullstop

**A11:** Done.

**R12:** 18,10-11: Close space between these two references

25 **A12:** Done.

**Reviewer #2**

**R13:** Abstract: Lines 24 and 25: remove one point.

**A13:** We removed the full stop. See also comment R2 and author response A2.

**R14:** Page 2, Line 14: authors may consider also:

Elosegi, A., Díez, J.R., Flores, L., Molinero, J., 2016. Pools, channel form, and sediment storage in wood-restored streams: Potential effects on downstream reservoirs. Geomorphology 279, 1–11. https://doi.org/10.1016/j.geomorph.2016.01.007

Gurnell, a. M., Sweet, R., 1998. The distribution of large woody debris accumulations and pools in relation to woodland stream management in a small, low-gradient stream. Earth Surf. Process. Landforms 23, 1101–1121. https://doi.org/10.1002/(SICI)1096-9837(199812)23:12<1101::AID-ESP935>3.0.CO;2-O

**A14:** We thank the reviewer for the literature suggestions. However, in the public discussion of this article, reviewer #3 made clear that the literature review in the introduction should focus on the scope of this study. Thus, we will not add these references to the manuscript.

**R15:** Page 2, lines 16-17: although is cited later, authors may consider here also:

Grabowski, R.C., Gurnell, A.M., Burgess-Gamble, L., England, J., Holland, D., Klaar, M.J., Morrissey, I., Uttley, C., Wharton, G., 2019. The current state of the use of large wood in river restoration and management. Water Environ. J. wej.12465. https://doi.org/10.1111/wej.12465

**A15:** We agree and added the reference to this section in the following way:

Original:
"Positive ecological impacts of LW on fish species (i.e. Kail et al., 2007; Roni et al., 2015) and the macro-invertebrate fauna (i.e. Seidel and Mutz, 2012; Pilotto et al., 2014; Roni et al., 2015) are documented.
Therefore, in stream restoration projects, the presence of large wood can result in rapid hydromorphological improvements (Kail et al., 2007)."

Modification:

"Positive ecological impacts of LW on fish species (i.e. Kail et al., 2007; Roni et al., 2015) and the macro-invertebrate fauna (i.e. Seidel and Mutz, 2012; Pilotto et al., 2014; Roni et al., 2015) are documented. A recent review of the hydromorphological and ecological effects of LW with focus on river restoration can be found in Grabowski et al. (2019).

In stream restoration projects, the presence of large wood can result in rapid hydromorphological improvements (Kail et al., 2007)."

**R16:** Page 2, line 24: examples of reference about large wood transport during floods can be find here:

Comiti, F., Lucía, A., Rickenmann, D., 2016. Large wood recruitment and transport during large floods : a review. Geomorphology, 23–39. https://doi.org/10.1016/j.geomorph.2016.06.016

Lucía, A., Schwientek, M., Eberle, J., Zarfl, C., 2017. Morpholocigal changes and large wood transport in two steep torrents during a severe flash flood in Braunsbach, Germany 2016. Sci. Total Environ. 20, 17920.

Mazzorana, B., Ruiz-Villanueva, V., Marchi, L., Cavalli, M., Gems, B., Gschnitzer, T., Mao, L., Iroumé, A., Valdebenito, G., 2018. Assessing and mitigating large wood-related hazards in mountain streams: recent approaches. J. Flood Risk Manag. https://doi.org/10.1111/jfr3.12316

Ruiz-Villanueva, V., Bodoque, J.M., Díez-Herrero, A., Bladé, E., 2014. Large wood transport as significant influence on flood risk in a mountain village. Nat. Hazards 74, 967–987. https://doi.org/10.1007/s11069-014-1222-4

Ruiz-Villanueva, V., Badoux, A., Rickenmann, D., Böckli, M., Schläfli, S., Steeb, N., Stoffel, M., Rickli, C., 2018. Impacts of a large flood along a mountain river basin: the importance of channel widening and estimating the large wood budget in the upper Emme River (Switzerland). Earth Surf. Dyn. 1–42. https://doi.org/10.5194/esurf-2018-44

Wohl, E.E., Bledsoe, B.P., Fausch, K.D., Kramer, N., Bestgen, K.R., Gooseff, M.N., 2016. Management of Large Wood in Streams: An Overview and Proposed Framework for Hazard Evaluation. Am. Water Resour. Assoc. 1482. https://doi.org/10.1017/CBO9781107415324.004

**A16:** We thank the reviewer for the literature suggestions. As already mentioned in author response A14 a previous reviewer stressed to focus on the scope of this study in the literature review. A more detailed review on large wood transport is beyond the scope of this study and may facilitates the distraction of readers. Thus, we will not add additional references about this topic to the manuscript.

**R17:** Page 4: here you may also consider:

Manga, M., Kirchner, J.W., 2000. Stress partitioning in streams by large woody debris 36, 2373–2379.

5    Hygelund, B., Manga, M., 2003. Field measurements of drag coefficients for model large woody debris. Geomorphology 51, 175–185. https://doi.org/10.1016/S0169-555X(02)00335-5

**A17:** We thank the reviewer for the literature suggestions. We are aware of both publications but haven't added them to previous versions of the manuscript as they (among other studies) focus on large wood effects on shear stress and the drag
10    coefficient. Instead, we focussed on reviewing studies concerning the effect of LW on the roughness coefficients of the Manning or Darcy-Weisbach equation (Manning's n or Darcy-Weisbach's f) which are standard calibration parameters in hydrodynamic models. As this study focusses on implementing large wood in a hydrodynamic model we find that information about the effects of large wood on i.e. shear stress may be an important fact that needs to be mentioned in designated review articles on the hydraulics of large wood but, in this case study, rather leads to the distraction of readers from the main topic.
15    Thus, we will not add these references to the manuscript.

**R18:** Page 4: Line 9: about backwater rise, please consider:

Schalko, I., Lageder, C., Schmocker, L., Weitbrecht, V., Boes, R.M., 2019. Laboratory Flume Experiments on the Formation
20    of Spanwise Large Wood Accumulations: Part I: Effect on backwater rise. Water Resour. Res. 0–3. https://doi.org/10.1029/2019WR024789

**A18:** We thank the reviewer for the literature suggestion. In this case and after careful studying the article, we agree that this very recent study should be mentioned as an example investigating the large wood induced backwater rise. Thus, we added
25    the reference in the following way:

Original:
"The large wood induced alteration of channel roughness coefficients and overall hydraulic impacts such as backwater effects are crucial for the identification of local risks."

Modification:
"The large wood induced alteration of channel roughness coefficients and overall hydraulic impacts such as backwater effects, for instance investigated by Schalko et al. (2019), are crucial for the identification of local risks."

**R19:** Page 4, line 25: replace were by where

**A19:** Done.

**R20:** Page 5, line 7: the bed material is not visible in Figure 2, please provide a description about the grain size, if data is not available.

**A20:** As stated in the scope of the public discussion, no data about grain sizes is available. The general grain size classes along the study reach are described in the section about the morphological characteristics of the study reach (chapter 2, page 5, lines 3-11).

**R21:** Figure 2: the LW pieces are not so visible in the pictures, maybe you can add arrows or try to highlight them. Some of them seems to be outside the channel…or hanging above…

**A21:** Figure 2 was added based on a previous reviewer comment in the public discussion to give the reader a better impression regarding the overall morphology of the channel and riparian areas. The photographs do not to show the large wood elements of the field experiments. As stated in the methods section (chapter 3.2, page 6, lines 21-28), the field experiments took place in 2008 with the experiment results already published elsewhere. In contrast, the photographs in figure 2 were taken in May 2017 due to the fact that photographs taken during the field experiments in 2008 are not sufficient to give a morphological overview of the reach. Although this information is already given in the caption of the figure, we clarified this in the following way:

Original caption:
"Figure 2: Detailed map of the study reach (topographic data outside reach: GeoSN, 2008). Photographs were taken in May 2017 in the direction of flow (north to south)."

Modified caption:
"Figure 2: Detailed map of the study reach (topographic data outside reach: GeoSN, 2008). Photographs were taken 9 years after the field experiments in May 2017 in the direction of flow (north to south)."

**R22:** Page 5, line 13: what about the effect of these boulders and channel spanning steps on the flow? How were the boulders and steps reproduced by the model? Also as roughness elements?

**A22:** Detailed information are already given in chapter 5.1, page 11, lines 19-31.

**R23:** Page 6, line 3: why this model produces a higher goodness-of-fit compared to others? Which other models have been compared with this one? The reasons given in the text to choose the model are not fully related to the conditions in the study reach (dike breaching…). In my opinion, there is no need to justify the choice of this model here, but the advantages and
5    limitations of the model, and a brief comparison with other available tools should be discussed in the discussion.

**A23:** Information about the hydrodynamic model used as well as a brief assessment of the suitability of the model to achieve the aims of this study were demanded by a previous reviewer and thus, will not be altered.

We do not argue that hydrodynamic models are the only tools available to estimate the effects of large wood. However, the
10   main topic of this study is to test and compare the implementation of large wood in a hydrodynamic model. Thus, an additional discussion and comparison with alternative tools, i.e. empirical models on backwater rise, is beyond the scope of this case study.

**R24:** Page 6, line 25: if logs were placed lengthwise, I guess the effect would be much lower than in the case of perpendicular
15   logs. Moreover, the height between the log and the river bed is important. If logs were standing above the channel bed (hanging from the river banks), water is flowing under and above the LW piece. This effect could not be reproduced by adding discrete obstacles in the calculation mesh, without porosity…please, discuss this issue more in detail. This is also the main reason why Keys and Hafs et al used a different approach…

A deeper discussion about the representation of the actual shape of LW pieces is also missing in the discussion. Although just
20   cited in the introduction, it could be useful to discuss this topic and consider the work by:

Allen, J.B., Smith, D.L., 2012. Characterizing the Impact of Geometric Simplification on Large Woody Debris Using CFD. Int. J. Hydraul. Eng. 1, 1–14. https://doi.org/10.5923/j.ijhe.20120102.01

**A24:** This is true. The effect of large wood elements depends i.e. on their orientation and also on their permeability in terms of water flowing through branches, underneath or over elements. Two-dimensional hydrodynamic models cannot simulate diffuse flow of water though elements resulting in non-permeable elements. This issue is likely to be responsible for the differences between simulation and observation of the results of variant V3 and is already discussed in detail in the discussion
30   section (chapter 5.3, page 13, lines 21-31).

Keys et al. (2018) used weir embankments where water flows through randomly placed orifices in the weir or over it. Hafs et al. (2014) used discrete wood elements not spanning the entire channel and hence, allowed water to pass at one side. In a similar way, water can flow past discrete elements in the present study by flowing through the unblocked channel and riparian

zones. Consequently, different types of representations for two-dimensional simulations exist and may be compared in future studies. We already discuss this in chapter 5.3, page 14, lines 3-17.

References addressing three-dimensional models or CFD were removed from the discussion based on the comments of previous reviewers, due to the reason that this study focusses on simulations with a two-dimensional model. A fully resolved 3-D model allows a diffuse water flow through large wood elements and hence, an element design much closer to the actual large wood geometry. In two-dimensional models, stronger simplifications are necessary and hence, the results of this study are difficult to compare with studies using three-dimensional models. Thus, we will not re-add and discuss studies using CFD in the discussion chapter.

**R25:** Page 7, line 9: please, provide some more details about the interpolation methods used here (although you provided the reference, it could be very useful for the readers to know what you did)

**A25:** We are glad about this comment. We re-checked the entire interpolation procedure and found that we did not make use of the interpolation algorithm after Hutchinson (1989) in the latest and final DTM processing approach. Instead, we created a triangular irregular network (TIN) which was then transformed into a raster and equally spaced elevation points. This was done to be consistent with the linear interpolation conducted between elevation profiles in the channel in the scope of the subsequent generation of the calculation mesh. We apologize for this mistake in previous versions of our manuscript, remove Hutchinson (1989) from the reference list and modify the section in the following way:

Original:

"The final DTM for the model is generated from processing and combining all topographic datasets in the software environment ArcGIS v10.5 (ESRI Inc., USA) using the implementation of the procedure described in Hutchinson (1989) for interpolation."

Modification:

"The final DTM for the model is generated from processing and combining all topographic datasets in the software environment ArcGIS v10.5 (ESRI Inc., USA) creating a triangular irregular network (TIN) before transforming it into a raster dataset. The resulting DTM is exported as equally spaced elevation points with a spatial resolution of approximately 0.5 x 0.5 m for the entire study reach including riparian areas and embankments."

**R26:** Page 10, line 30 and elsewhere: remove references from the results section

**A26:** Here, the reference is necessary as we refer to the classification of goodness-of-fit parameters published by Moriasi et al. (2007).

**R27:** Page 12, line 15: however, the area affected by the presence of LW has not been analysed here, and V1 assumes an increase in the roughness along the entire reach, right?

**A27:** This is correct.

**R28:** Page 12, lines 20-28: I am not sure why the roughness was not adjusted along the floodplain during the calibration without LW. This is also important in terms of the storage of water and the resulted hydrograph downstream, as described in the results…I would suggest providing results with the calibrated roughness in the floodplain only. And, how many values of roughness values have been used? Did you map homogeneous roughness units? Or just assumed one single value for the entire domain? Please, provide details and justify. A map with roughness values could help, and a discussion about that would be beneficial.

**A28:** The channel and floodplain roughness coefficients were iteratively calibrated to achieve the best simulation result in the reference variant. This information is already given in chapter 3.4, page 8, lines 12-13. The calibrated roughness value for riparian area in the base/reference variant is given in the results section (chapter 4.1, page 9, line 13).

As a previous reviewer stated, it would be more rigorous to not to change the calibrated baseline roughness value of the floodplain in variants V1 and V2. Therefore, we think it is important to describe and show the results with and without subsequent adjustment of roughness coefficients in the riparian area.

We estimated roughness coefficients for channel sections and the floodplain in the field as stated in the methods chapter (chapter 3.2, page 7, lines 20-23). For the floodplain we chose a single value due to the mostly grassy vegetation cover while different roughness values were assigned to channel sections. To make this clearer, we changed the sentence in the following way:

Original:

"The Strickler coefficients $k_{st}$ were estimated during field surveys in May 2017 with reference to established roughness coefficient classifications for different land cover and surface material types (i.e. Chow, 1959) as well as in accordance with observed ground cover during field experiments in 2008."

Modification:

"The Strickler coefficients $k_{st}$ were estimated for channel sections with similar bed material and the floodplain during field surveys in May 2017 with reference to established roughness coefficient classifications for different land cover and surface material types (i.e. Chow, 1959) as well as in accordance with observed ground cover during field experiments in 2008."

**R29:** Page 12, line 30 and following lines: this part of the discussion is very interesting, maybe a table summarizing these studies and their main findings together with the results from this work could be very useful for the readers.

**A29:** Although we agree that a more detailed review of existing studies on the effects of LW on roughness coefficients with varying boundary conditions would be interesting, it is beyond the scope of this case study. However, it may be considered in designated review articles addressing the hydraulics of large wood.

**R30:** Figure 3: the evaluation and comparison between simulations and observations is missing in the workflow.

**A30:** We agree and changed the figure in the following way:

Original:

[Figure]

Modification:

[Figure]

**R31:** Figure 4: some improvements could be useful. Please, add a scale, flow direction, and also arrows showing the LW pieces could be useful. Actually, it could be better to show the roughness maps instead of just the calculation mesh.

**A31:** We agree and added a scale indicator to one of the LW elements as this is a three-dimensional figure where the scale varies throughout the image. In addition, flow direction and arrows pointing to the large wood pieces were added. Moreover, we added another map showing the distribution of roughness coefficients in the study reach and adjusted the manuscript accordingly.

[Figure]

Modification:

[Figure]

New figure:

[Figure]

**Figure 6: Distribution of calibrated roughness coefficients of all simulation variants with and without LW in the study reach: a) reference variant RV without LW, b) variant V1 with stable LW as an increase of roughness in the entire channel, c) variant V2 with stable LW as an increase of roughness at element positions only and, d) variant V3 with LW as discrete topographic elements of the calculation mesh. If displayed, values in brackets represent the Strickler coefficient after subsequent adjustment.**

**R32:** Figure 5: adding inflow and outflow (in parenthesis together with the corresponding weir) could be helpful.

**A32:** We modified the figure in the following way:

Original:

[Figure]

Modification:

[Figure]

Mean observed hydrographs with and without LW during field experiments

**Complete reworked manuscript with all modifications shown:**

[revised manuscript text omitted]

a)

b)

a)

Direction of flow

b)

LW elements

7.8 m

**Figure 4: a) Calculation mesh of the hydrodynamic model used in simulation variants RV, V1 and V2 with the use of variable Strickler coefficients adjusted for the entire channel (V1) or adjusted at the positions of all LW elements only (V2) and b) mesh with discrete LW elements used in variant V3. Example of the first 60 m of the study reach.**

[Figure]

[Figure]

**Mean observed hydrographs with and without LW during field experiments**

Legend:
- Thomson-weir 1 without LW (corrected)
- Thomson-weir 1 with LW (corrected)
- Thomson-weir 2 without LW
- Thomson-weir 2 with LW

Y-axis: Discharge [ m³ s⁻¹ ]

X-axis: Experimental run time [s]

[Figure]

**Figure 5:** **Average measured and corrected flood hydrographs observed during field experiments with and without stable in-channel large wood at both Thomson-weirs (after Wenzel et al., 2014).**

[Figure]

**Figure 6: Distribution of calibrated roughness coefficients of all simulation variants with and without LW in the study reach: a) reference variant RV without LW, b) variant V1 with stable LW as an increase of roughness in the entire channel, c) variant V2**

[Figure]

**Figure 7̶6: Best simulated mean flood hydrographs of all simulation variants with and without LW at Thomson-weir 2: a) results of
10    the reference variant RV without LW, b) variant V1 with stable LW as an increase of roughness in the entire channel, c) variant V2 with stable LW as an increase of roughness at element positions only and, d) variant V3 with LW as discrete topographic elements of the calculation mesh. For simulation variants V1 and V2 the best fit with and without subsequent adjustment of riparian Strickler coefficients is displayed. The Nash-Sutcliffe-Efficiency (NSE) is shown for each simulation variant. If displayed, values in brackets represent the NSE of simulations without adjustment of riparian roughness coefficients.**

**Table 1: Average observed and simulated discharge sums (m³ h⁻¹) at both Thomson-weirs for all simulation variants. For variant V1 and V2 discharge sums with subsequent adjustment of riparian Strickler coefficients are displayed.**

| Discharge sums (3600 s) for each variant (m³ h⁻¹) | Base-Variant | Variant 1 | Variant 2 | Variant 3 |
|---|---|---|---|---|
| Thomson-weir 1 (observed, corrected) | 128 | 128 | 128 | 128 |
| Thomson- weir 2 (observed) | 132 | 133 | 133 | 133 |
| Thomson-weir 1 (simulated) | 128 | 128 | 128 | 128 |
| Thomson-weir 2 (simulated) | 128 | 128 | 128 | 123 |
| Difference between observed and simulated values (Thomson-weir 2) | -4 | -5 | -5 | -10 |
| Observed difference between Thomson-weir 1 and 2 | -4 | -5 | -5 | -5 |

**Table 2: Calculated statistical goodness-of-fit parameters for all simulation variants. For variant V1 and V2 goodness-of-fit parameters with and without subsequent adjustment of riparian Strickler coefficients are displayed.**

| Goodness-of-fit parameters | Basie-Variant | Variant 1 without adjustment | Variant 1 | Variant 2 without adjustment | Variant 2 | Variant 3 |
|---|---|---|---|---|---|---|
| NSE | 0.99 | 0.97 | 0.98 | 0.94 | 0.96 | 0.90 |
| RSR | 0.11 | 0.18 | 0.14 | 0.24 | 0.19 | 0.32 |
| PBIAS (%) | - 3.5 | - 3.6 | - 3.7 | - 4.2 | - 4.0 | - 7.7 |

5  **Table 3: Attributes of approaches for large wood implementation applied in this study relative to the reference variant without large wood. Signs indicate an attribute being higher (+), lower (-) or equal (o) to the simulation without stable large wood.**

| Attribute | Variant V1 – reach-wise increase of roughness | Variant V2 – section-wise increase of roughness | Variant V3 – large wood as discrete elements |
|---|---|---|---|
| Work and time consumption | + | ++ | ++++ |
| Computational time | o | o | + |
| Statistical goodness-of-fit | - | -- | --- |
| Visual goodness-of-fit (hydrograph shape) | -- | -- | - |